# Synchronization in electric power networks with inherent heterogeneity up to 100% inverter-based renewable generation

Amirhossein Sajadi [1✉], Rick Wallace Kenyon[1,2,3] & Bri-Mathias Hodge[1,2,3]

The synchronized operation of power generators is the foundation of electric power network stability and a key to the prevention of undesired power outages and blackouts. Here, we derive the conditions that guarantee synchronization in power networks with inherent generator heterogeneity when subjected to small perturbations, and perform a parametric sensitivity analysis to understand synchronization with varied types of generators. As inverter-based resources, which are the primary interfacing technology for many renewable sources of energy, have supplanted synchronous generators in ever growing numbers, the center of attention on associated integration challenges have resided primarily on the role of declining system inertia. Our results instead highlight the critical role of generator damping in achieving a stable state of synchronization. Additionally, we report the feasibility of operating interconnected electric grids with up to 100% power contribution from inverter-based renewable generation technologies. Our study has important implications as it sets the basis for the development of advanced control architectures and grid optimization methods that ensure synchronization and further pave the path towards the decarbonization of the electric power sector.

[1] Renewable and Sustainable Energy Institute (RASEI) at the University of Colorado Boulder, 4001 Discovery Drive, Boulder, CO 80303, USA. [2] Electrical, Computer, and Energy Engineering (ECEE) Department at the University of Colorado Boulder, 425 UCB, Boulder, CO 80309, USA. [3] Grid Planning and Analysis Center at the National Renewable Energy Laboratory (NREL), 15013 Denver W Pkwy, Golden, CO 80401, USA. ✉email: Amir.Sajadi@colorado.edu

The decarbonization of power networks is an ongoing global effort that is rapidly accelerating with the growing recognition that it is an essential keystone to achieve a sustainable energy future[1]. Electric power network decarbonization will require the large-scale deployment of carbon-free technologies, with variable renewable power generation expected to increase greatly. Among the variable renewable power generation technologies, solar photovoltaics (PV) and wind power plants are now cost competitive with conventional generation in most locations and the cost of energy production using renewable power plants continues to decline[2]. Accordingly, it is anticipated that variable renewable generation technologies will continue to dominate the new installed generation capacity over the next two decades, spurring a transition to 100% renewable-based power networks with a substantially altered landscape for the associated planning, management, stability, and control approaches[3–5]. This transition is already underway and accelerating quickly. In 2020, solar and wind resources accounted for more than 72% of all new electricity generation capacity globally[6]. This set a new record for the expansion of renewable installations by more than 45% from 2019[7]. In the US alone, a new record was set in 2020 for renewable expansion as a total of 80% of new electricity generation capacity installed came from solar and wind resources, with solar accounting for 43%[8] and 37% for wind[9] of all new generation capacity installed. It is expected that solar and wind will continue to break deployment records around the globe, with renewable resources anticipated to account for approximately 90% of new generation capacity in 2021 and 2022[10]. In the US alone, it is anticipated that 70% of new capacity installed in 2021 will be solar and wind power plants[11].

One of the main challenges pertaining to the integration of variable renewable energy resources into power networks is that they are integrated with a power electronic interface known as an inverter; they are therefore commonly referred to as inverter-based resources (IBRs). This stands in contrast to most conventional power plants where electricity is generated using synchronous generators. These synchronous generators are being supplanted by IBRs, and as a result, many of the fundamental assumptions that provide the foundation for the contemporary maintenance of power network stability, approaches based on the characteristics of synchronous generators, may no longer be valid for power networks with very high levels of IBRs. This shift is due to the associated paucity of synchronous generators and increased heterogeneity of constituting parameters across the networks of this class as a result of the adaptive inertia and damping that the IBR offer. In this paper, we study synchronization in electric power networks with high levels of IBR generation, at levels up to 100%. This is a dynamic problem, with foundations to the stable operation of interconnected power network, whose impetus is to understand whether a network remains stable following a disturbance.

The problem of synchronization in power networks aims to assess frequency dynamics and identify the necessary conditions and mechanisms for a network to maintain synchronization. In power networks, the coupling variable is the device frequency; when synchronized and in steady state, this value will be consistent across the network. Each generator exhibits a frequency that evolves according to power export; this is commonly known as frequency response. This problem is growing in popularity and has attracted many scholars in the physics and applied mathematics fields. Among the existing body of literature, we identify several approximations of the power network as nonlinear oscillators including Kuramoto oscillators[12–17], Lienard oscillators[18–20], and Van der Pol oscillators[21]. For oscillator approximations, several critical factors are simplified and neglected so that the governing equations of motion (commonly known as the swing equation) can resemble the oscillator of interest. We also recognize more comprehensive studies in which closed-form solutions have been developed[22,23]. The former study[22] offers provisional conditions and the latter[23] assumes the homogeneity of coupling damping factors, which not only falls short to emulate the intrinsic heterogeneous characteristic of real-world power networks presently dominated by synchronous generators, but is especially lacking for future networks potentially dominated by renewable energy technologies.

In this paper, we advance the state of the art concerning the frequency synchronization in modern power networks involving three contributions. First, we introduce heterogeneous coupling factors that are inherent in real-world power networks and derive the necessary and sufficient conditions for synchronization in power networks (i.e., at least one generator with a frequency forming relationship and all generators with positive damping coefficients); the current state of the art resides at homogeneous coupling factors[23]. The consideration of heterogeneous coupling damping factors allows us to study different configurations and combinations of conventional and emerging renewable generation technologies, enabling us to more directly address the challenge of decarbonization. The dynamics of frequency response of power generation devices can be characterized by a second-order ordinary differential equation (ODE). Therefore, we model generators as second-order dynamic systems in the form of $\frac{d^2x}{dt^2} + 2\zeta\omega_n\frac{dx}{dt} + \omega_n^2x = 0$ with the critical parameters of natural frequency, $\omega_n$, and damping ratio, $\zeta$, that constitute the response agility, oscillations, and steady-state, implicitly represent all contributing components to the inertia and effective damping, e.g., inertial momentum, droop control, etc. Second, we characterize power network synchronization parametrically and develop a mechanism for its enhancement by adjusting the key contributing generator parameters. The findings of our parametric sensitivity analysis highlight the significance of the damping component and derive a concise relationship of its impact in achieving a stable synchronized state (a matter alluded to and generally identified in the recent literature[24–26] without a concise identification of the extent of its contribution) and the benefits that in fact reduced inertia component offers, which may be intuitive to scholars versed in dynamical systems and complex network science. Our analysis, which is based on the dynamic response of second-order systems, offers a simplified yet accurate model to capture frequency dynamics with both synchronous generators and inverter-based resources. They are efficient and well-applicable to frequency response studies in large-scale power networks and our extensive numerical experiments validate them. Third, after establishing the new formalism, we leverage it and demonstrate the feasibility of operating electric power networks up to 100% inverter-based energy technologies with enhanced synchronization capabilities, provided sufficient damping element in the form of fast responding headroom power reserve.

We are optimistic that the findings in this paper provide a new perspective on power networks with high shares of renewable technologies. Currently, this field is known as *low-inertia power systems*[27]. This new perspective may help scholars versed in electric power systems to shift their attention from the concerns stemming from reduced inertia towards the benefits it offers, given the capabilities of grid-forming inverters to stabilize the system with the addition of a substantial damping component.

## Results

**Synchronization in electric power networks**. Electric power networks are one of the most complex dynamical networks ever engineered, and ensuring their synchronization is pivotal because the lack thereof may result in any disturbance yielding sustained

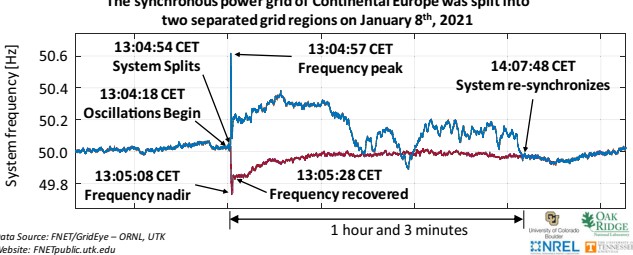

**Fig. 1 Frequency traces of the European synchronized continental power network recorded by Phasor Measurement Units across this network on January 8, 2021.** In this plot, two distinct frequencies are visible after a loss of synchronization when the network was split into two sub-networks with an interruption in power delivery for a duration of 1 h and 3 min.

oscillations and a loss of stability[28]. Synchronization in a power network can be interpreted as a stable state when the pace of evolution of the electric angle in all generators across the network is identical; in a power network with $n$ generators, it can be mathematically described by:

$$\dot{\delta}_1 = \dot{\delta}_2 = \cdots = \dot{\delta}_n = \omega_{\text{sync}} \tag{1}$$

where $\delta_i$ represents the position of the electric angle of the $i$th generator and $\omega_{\text{sync}}$ is the synchronized speed. The dot notation indicates the time derivative; $\dot{x} = \frac{d}{dt}x$. The synchronized operation of generators guarantees that the flow of electric power across the network remains stable and yields a homogeneous frequency at all nodes; i.e., a network frequency ($f \approx (2\pi)^{-1}\omega_{\text{sync}}$). Instabilities have been observed in power networks synchronized frequency on several occasions over the past half century leading to an interruption of power delivery[29–34]. Most recently, the European continental power network experienced a loss of synchronization on January 8, 2021[34], as shown in Fig. 1.

A mechanical analog of the synchronized operation of two generators in an alternating current (AC) electric power network, such as the small system shown in Fig. 2a, is depicted in Fig. 2b. Both networks are damped oscillators with two degrees of freedom. When subjected to an external force, i.e., a small disturbance, the equilibria for these networks is the local minima of the corresponding energy functions; for the mechanical network it is $\sum_{i=1}^{2} 0.5\, m_i \cdot v_i^2$ where $v_i$ is the mechanical speed of $m_i$ which is $i$th body of mass, and for the electrical network it is $\sum_{i=1}^{2} 0.5\, M_i \cdot \omega_i^2$ where $\omega_i = 2\pi f_i$ is the electric angular speed and an electric frequency approximate, $f_i$, of $i$th generator with $M_i$ inertia coefficient (Supplementary Note 1). Suppose two bodies of mass are located on two wheels with negligible friction with the surface upon which they have freedom of movement in the $x$ direction. When the mechanical network is subjected to an external force (which is the mechanical analog of a change in loading conditions in an electric power network), the position of the bodies of mass (which are the mechanical analogs of electric angle in AC networks) and subsequently the distance between the two bodies, $(x_1 - x_2)$, may change. This change of in the distance between the two bodies results in force exerted by the spring, described by $k(x_1 - x_2)$ where $k$ is the spring constant. This force is directly the mechanical analog to the DC approximation[35] of changes of flow of electric power on an assumed lossless transmission line that connects the two generators in the electric network, expressed by $Y(\delta_1 - \delta_2)$, where $Y$ is the admittance of the line and $\delta_i$ is the position of the electric angle in $i$th generator, assuming the voltage at the two terminating busbars is unity. Accordingly, the stable synchronized operation in the mechanical network is defined as the ability to damp out the transient forces induced in the spring following a small perturbation, and regain a

steady distance between the two bodies of mass which manifests itself in the form of displacement of the two bodies with identical speeds. Figures 2c through 2f show the results for the synchronized speed of the two bodies following a small disturbance with varied parameters. The results here suggest that mechanical networks with smaller inertial mass and greater damping characteristics possess a more robust and stable natural synchronization capability. This notion is intuitive as although the mass provides an inertial resistance to the acceleration of a body when subjected to force, it also creates an inertial resistance to slowing down the mass once moving. In addition, the higher damping capability allows the network to more effectively dissipate the kinetic energy produced by an external force. The results presented in Figs. 2g through 2j support this intuition as (1) the amount of kinetic energy induced in the mechanical network is directly proportional to the amount of inertial mass; the smaller the mass the smaller the induced kinetic energy to dissipate, and (2) the ability to dissipate the kinetic energy is proportional to the damping coefficient; the larger the dampers, the faster the kinetic energy is dissipated.

The results presented here underscore the pivotal importance of studying the problem of synchronization in interconnected networks under the assumption of heterogeneous inertial and damping coefficients in order to fully capture the complex network dynamics. The assumption of homogeneity in the characteristics of subsystems, subnetworks, or components of an interconnected network is rarely valid for real-world large-scale networks; embedding such assumptions into an analytical model to study real-world, complex, interconnected networks may distort the findings. The results here yield that for the cases that involved homogeneous inertial and damping coefficients for subsystems/subnetworks, the numbers of oscillatory modes that are observed are limited relative to those observed in the cases that involved heterogeneous coefficients. Hence, the assumption of homogeneity of inertial and damping coefficients limited the understanding of a complete range of critical modes in this study.

The analogies between the two models suggest the existence of a speed-frequency analogy to translate the concepts, contributing factors, and solutions pertinent to mechanical networks into those of the electric networks. Therefore, as a natural next step, the remainder of this paper applies the intuition of synchronization in mechanical networks with emphasized heterogeneity impacts to a mathematical description and analytical derivation for synchronization in electric power networks.

**Dynamic model of power network.** Electric power networks are a class of complex dynamic networks with power generators, substations, and load centers constituting the nodes, and the interconnecting transmission lines constituting the links. This study focuses on the impacts of generator parameters on the network synchronization, manifested in the form of frequency heterogeneity across the network. Therefore, generators can be modeled as dynamic elements while the transmission lines and consuming loads are treated as algebraic elements[36,37]. We consider three power generation technologies that are the primary options for transitioning to 100% renewable-based power networks. The first technology is the synchronous generator; an overwhelming majority of existing generation facilities are equipped with this technology including hydro, nuclear, natural gas, and coal-fueled power plants[38,39]. The second and third technologies are based on power electronic inverters, which interface variable renewable energy sources and energy storage units with the grid. They can be categorized as (2) grid-following inverters (referred to as GFL, henceforth) and (3) grid-forming inverters (referred to as GFM, henceforth). The GFL is the most

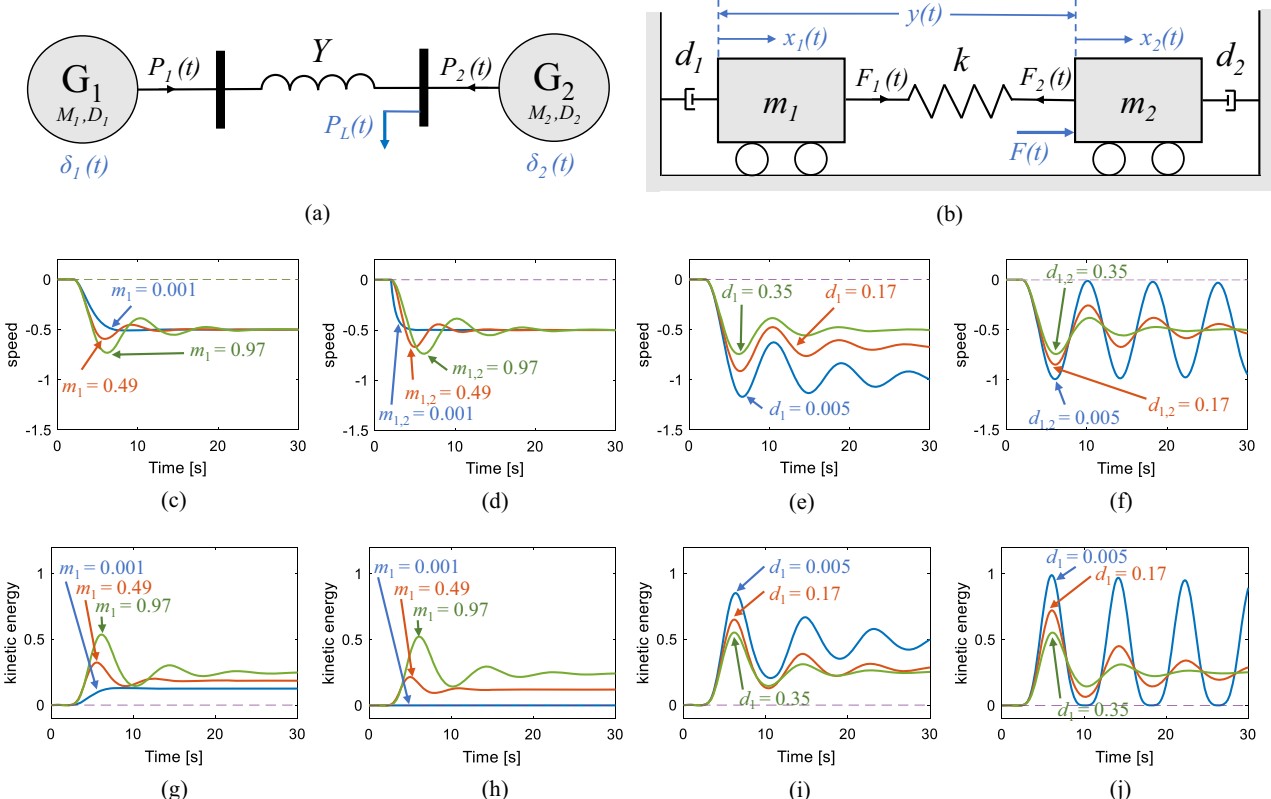

**Fig. 2 Mechanical analog of synchronized operation of generators in an electric power network. a** An electrical power network with two generators, $G_1$ and $G_2$, whose inertia and damping coefficients are $M_1$, $M_2$, $D_1$, and $D_2$, respectively. The two generators are interconnected through a transmission line with an admittance value of $Y$ and respond to the changes of demand of electric load $P_L$. **b** A mechanical analog of the electrical power network with two bodies of mass $m_1$ and $m_2$ tied to their nearest wall with dampers $d_1$ and $d_2$, respectively. The bodies of mass are interconnected via a spring with constant of $k$ and respond to the external force $F$. Default values for constants in this network are $m_1 = m_2 = 1.00$, $d_1 = d_2 = 0.35$, and $k = 0.30$. The mechanical network here is subjected to an external force $F$ and while the default values are kept unchanged, only one or two values are varied in order to conduct a parametric sensitivity analysis of the response. **c, g** Dynamics of speed and kinetic energy for varying values of $m_1$, heterogeneous inertial mass condition.
**d, h** Dynamics of speed and kinetic energy for varying values of $m_1$ and $m_2$, homogeneous inertial mass condition. **e, i** Dynamics of speed and kinetic energy for varying values of $d_1$, heterogeneous damping condition. **f, j** Dynamics of speed and kinetic energy for varying values of $d_1$ and $d_2$, homogeneous damping condition. These results indicate that mechanical networks with a smaller inertial mass (all Blue traces in **c, d, g, h**) and greater damping characteristics (all Green traces in **e, f, i, j**) offer a more robust and stable natural synchronization capability, as the speed reaches an equilibrium faster and the kinetic energy is dissipated quicker.

common currently used class of inverters, while the GFM as applied in parallel on power systems an emerging and promising technology. The analysis of synchronization in power networks considers only the dynamics associated with frequency response[23,40]. The generic equations of motion for both the synchronous generator and the GFM can describe the evolution of the electric angle, $\delta$, as given by:

$$\ddot{\delta} = M^{-1}(p^* - p_e - D\dot{\delta}) \qquad (2)$$

where $p^*$ and $p_e$ are the desired power output and actual power output, respectively. The desired power output for the synchronous generator is equivalent to the mechanical power whereas in the GFM it is the power set point. The $M$ and $D$ are inertia and damping coefficients, respectively, for the synchronous generator and the GFM inverter. The inertia coefficient of a device implicitly represents all of its components contributing to the effective inertia that determines the pace at which the frequency deviates. In a synchronous generator, it is directly the mechanical inertia, whilst in the GFM model, it is a function of the cutoff frequency of the power measurement low-pass filter and the droop gain. Similarly, the damping coefficient of a device implicitly represents all of the components contributing to the effective damping that directly dictate its ability to dissipate the transient frequency

oscillations and regain an equilibrium. For the synchronous generator, it involves the product of the mechanical inertia and droop gain, the response time of turbine and governor, and the damping torque provided by the damper windings. In the GFM model, it involves the cutoff frequency of the power measurement low-pass filter and the droop gain. See Supplementary Note 2 for the explicit model description and validation of this generic second-order model. The chief contrast between the two technologies is that in a synchronous generator the inertia and damping coefficients are constants dependent on the machine design, and can be expressed by the $M^{-1}D$ ratio of a generator for simplicity[38,40] whereas in the GFM, the inertia and damping coefficients can be adaptively adjusted. This is because power electronic inverters have kHz switches driven by digital controllers which allows the adaptive change of control loop gains that can exhibit dynamic responses with decoupled damping and inertia components, granted an available headroom power reserve[41,42]. The GFM technologies can be generally categorized into virtual synchronous machine (VSM) and multi-loop GFM. Both technologies have a frequency response with dynamics that can be described by a second-order differential equation. Therefore, the GFM model in this paper represents both technologies; when $M \gg 0$ the VSM control is employed[43] and for $\{M \approx 0 \wedge$

$M \neq 0$} the control mode is the multi-loop droop class of GFM[25]. The GFL, on the other hand, simply follows the grid frequency measured at the point of interconnection, using the estimation provided by a phase-locked loop (PLL). Therefore, the governing equation is given by:

$$0 = p^* - p_e \tag{3}$$

where $p^*$ and $p_e$ are the power setpoint and power export (where the power setpoint applies to the point of measurement of power exported and internal dynamics are implicitly embedded). Effectively, the GFL can be modeled as a negative constant load because of its absence in the construction of frequency. The detailed discussion about the generators and their explicit models are provided in Supplementary Note 2.

The generators are interconnected to substations and load centers by a time-invariant, electrical, mesh network of power lines commonly known as the transmission network. The lines are electric circuits whose parameters, that assumed constant here because the timescale of interest, which is cycles to seconds[23,38,40], consist of an admittance (see the "Methods" section). Individual line admittance and node connection criteria is used to form the network admittance matrix, which is a square and symmetric matrix that describes the network topology (see "Methods"). Next, we find the steady-state equilibrium of the network using the admittance matrix and by solving the network power-flow equations, which determines the power dispatch of generators, line transfer quantities, and the voltage magnitudes and angles of all nodes (see "Methods"). Given the cycles to seconds timescale of interest in this study, it is assumed that the load is constant (i.e., there are no variations in networking loading besides explicit perturbations). Once the steady state equilibrium is determined, we use Kron reduction to acquire a lower dimensional electrical-equivalent network[44]. This is achieved by keeping only the nodes with a dynamic element directly interconnected and eliminating all algebraic elements including substations, loads, and GFL inverter-backed generators through algebraic manipulation (see "Methods").

For the assessment of the network synchronization when subjected to small perturbations (representing frequent events such as load fluctuations, regulator switching, and generation dispatch changes), we linearize the model (see "Methods") and appoint one generator as the reference angle generator[38,40]; all other angles are analyzed relative to the angle of the reference generator, $\delta_{i,n}$. This model is explicitly described and expanded in Supplementary Note 3 and the explicit details of coordinate transformation procedure follows in Supplementary Note 4. After a coordinate transformation, the network dynamics can be described as:

$$\begin{bmatrix} \Delta \dot{\delta}_{i,n} \\ \Delta \dot{\omega}_i \\ \Delta \dot{\omega}_n \end{bmatrix} = \begin{bmatrix} 0 & I & -1 \\ h_i & d_i & 0 \\ h_n & 0 & d_n \end{bmatrix} \begin{bmatrix} \Delta \delta_{i,n} \\ \Delta \omega_i \\ \Delta \omega_n \end{bmatrix} \tag{4}$$

where $\Delta$ is the linear difference operator, $\delta_{i,n}$ is the vector of $(n-1)$ relative electric angles and $\omega_i$ and $\omega_n$ are absolute electric speeds expressed by a vector of $(n-1)$ and an integer value (making it a vector with a single array), respectively. [0] is a matrix of zeros, and [I] is the identity matrix, each a square matrix of dimension $(n-1)$, and [−1] is a matrix with dimension $1 \times (n-1)$ and all elements are (−1). $h_i = -M_i^{-1} H_{i,j}$ and $d_i = -M_i^{-1} D_i$, $\forall i,j = 1, 2, \cdots, (n-1)$ are diagonal matrices, each a square matrix with dimension of $(n-1)$, with $H$, $D$, and $M$ the network interconnection Laplacian coefficients and generators damping and inertia coefficients, respectively. Finally, $h_n =$ $-M_n^{-1} H_{i,n}, \forall i = 1, 2, \cdots, (n-1)$ is a $1 \times (n-1)$ matrix and $d_n = -M_n^{-1} D_n$ is a single array.

**Conditions of stability**. We define a characteristic matrix for the network described in (4) as:

$$\begin{aligned}
p(\lambda) &= \alpha \lambda^3 + \beta \lambda^2 + \gamma \lambda + \xi = 0 \\
\alpha &= I \\
\beta &= -d_i - d_n I \\
\gamma &= (d_i d_n - h_i + [\mathbf{1} \otimes h_n]) \\
\xi &= (h_i d_n - d_i \cdot [\mathbf{1} \otimes h_n])
\end{aligned} \tag{5}$$

where the $\lambda's$ are the eigenvalues of the network and $\cdot$ and $\otimes$ are the inner and outer vector product operators. $\mathbf{1}$ is a matrix of all ones with dimension of $(n-1) \times 1$. The eigenvalues of this characteristic matrix are defined as the values subtracted from the diagonal elements that yield a singular matrix and thus $det(p(\lambda)) = 0$ as explicitly described in Supplementary Note 5. This condition yields a total of $(2n-1)$ eigenvalues whose eigenvectors are linearly independent for a network with $n$ generators, and a stable equilibrium exists if and only if all roots of the polynomial $p(\lambda)$ possess non-positive real parts (non-positive Lyapunov exponents). This condition yields the stability limit for synchronization is $|C| > 0$, $C = \{\exists M_i | M_i > 0, \forall i = 1, 2, \cdots, n\}$ and $D_i > 0$, $\forall i = 1, 2, \cdots, n$, implying that (i) the minimum of one non-GFL generator interconnected to the network is necessary to constitute the synchronization frequency and (ii) the damping coefficients of all generators must be positive. The stability analysis of the standard benchmark power network with 3-generators (also known as 9-bus test network)[38] is provided in Supplementary Note 6, as an example.

**Parametric analysis of synchronization**. The formal expression of the solution for Eq. (5) can be written in the form of

$$p(\lambda) = \prod_{i=1}^{(n-1)} \underbrace{\left( \lambda^2 + 2\zeta_i \omega_{n_i} \lambda + \omega_{n_i}^2 \right)}_{\text{internalmodes}} \cdot \underbrace{(\lambda + k_d)}_{\text{couplingmode}} = 0 \tag{6}$$

The first part represents the generator internal modes involving the electric angle and speed, which constitutes of a pair of complex conjugate eigenvalues. The second part is the network coupling mode that yields a real eigenvalue whose value is a function of the generator damping coefficient. Recalling Eq. (4), let us suppose a special condition where the damping factors, $d_i = -\frac{D_i}{M_i}$ are homogeneous i.e., $d = d_1 = d_2 = \cdots = d_n$. The stability criteria for this special condition was developed by Motter[23] and Machowski[40]. With this assumption, we can approximate internal modes, given that the dimension associated with the coupling mode can be reduced, as explained in Supplementary Note 7. Borrowing from Motter[23] and Machowski[40], the roots of the characteristic Eq. (5) are determined as:

$$\lambda = 0.5 d \pm 0.5 \sqrt{d^2 + 4\lambda_{h_i}} \tag{7}$$

where $\lambda_{h_i}$ is the eigenvalue of the submatrix $h_i$. Eq. (7) yields two internal modes for each generator that are complex conjugates in a stable network and pertain to the relative electric angle and relative speed with respect to those of the reference generator. The location of these eigenvalues determines the stability of the network.

Equation (7) suggests that the real part of these modes is proportional to the generator damping coefficient and thus, a reduction in the generator damping coefficient, $D$, directly moves the eigenvalues to the right closer to the $y$-axis, while an increase

directly moves them to the left further away from the y-axis. Both the generator damping factor $d$ and the network interconnection Laplacian $h$ have an inverse relationship with the inertia coefficient $M$, given $d = -\frac{D_i}{M_i}$ and $h_{ik} = -\frac{H_{ik}}{M_i}$. Therefore, for a stable network, a reduction in generator inertia increases the values of both the real and the imaginary terms and results in a migration further away from the origin to left-hand side. A concurrent reduction in generator inertia and damping coefficients increases the imaginary term whilst the real part, $d = -\frac{D_i}{M_i}$, remains unaffected because the numerator and denominator change at an identical rate. On the other hand, a reduction of the generator inertia coefficient and/or an increase of the damping coefficient will move the eigenvalues to the left-hand side of the plane.

Now that the internal modes are established, we assume that damping factors are heterogeneous and, therefore, $k_d \neq 0$, as the main distinguishing feature of our formalism from the existing body of literature[23,40]. This assumption brings the model more in line with the nature of power grids, where damping factors are heterogeneous because they depend on generation portfolios that encompass diverse sources and technologies for electricity generation with different damping characteristics. Under the heterogeneity assumption, there will be an additional real mode (the coupling mode described in (6)) and it is a function of the generator damping $D_i$ coefficient and the inverse of the inertia coefficient $M_i^{-1}$, as $d_i = -\frac{D_i}{M_i}$, and therefore directly migrates as a function of changes in these parameters. Induced synchronization frequency oscillations, caused when a system is subjected to a step disturbance, are primarily characterized by the complex eigenvalues with natural frequency of oscillations described as $\omega_n = \sqrt{\frac{1}{M_i}}$ and natural damping ratio described as $\zeta = \frac{D_i}{2\sqrt{M_i}}$ (see "Methods"). These relationships show that the natural frequency of oscillations, $\omega_n$, is independent of the generator damping coefficient, $D_i$, while the reciprocal square root of the inertia coefficient, $M_i$, determines the frequency of oscillations; the smaller the inertia coefficient, the faster the pace of frequency deviation and the higher the natural frequency of oscillations, but a reduced magnitude of the transient envelope and a shorter settling time. They also yield that the natural damping ratio, $\zeta$, is a function of the generator damping coefficient $D_i$, with a direct relationship, and the inertia coefficient, $M_i$, a reciprocal square root relationship; the higher the damping coefficient and the lower the inertia coefficient, the less deviant the frequency oscillations and, thus, shorter the settling time and reduced the magnitude of transient envelope.

The results from the 3-generator benchmark[38] corroborate these relationships and are presented in Figs. 3 and 4 and Table 1.

Our results establish the mechanism of synchronization in power networks and demonstrate that the stability can be enhanced by adjusting generator parameters such as the damping and inertia coefficients. Over the past few years, the displacement of synchronous generators has raised concerns over the potential ramifications on network stability, which has been widely characterized as a discussion about the impacts of reduced inertia, commonly known as *low-inertia power systems*[5,27,45,46]. Our results establish that the generator damping capability is more significant in achieving a stable state than the inertia. They also indicate the reduced inertia alone does not necessarily deteriorate dynamic performance when responding to small perturbations. In addition, our results demonstrate that multi-loop droop GFM inverter technologies with the capability to provide damping support independent of the inertial contribution[25] offer a great promise to enhanced grid stability.

**Application in modern power networks**. The synchronization conditions and mechanisms suggest that the stability of these networks is a function of (1) the generator parameters and (2) the interconnected network conditions. Accordingly, we tested 6 different complex power network benchmarks and carried out computer simulations for 1000 random parameters and conditions for each benchmark. We used the hertz-sec metric[47] to quantify the frequency response (see "Methods"). This metric is a proxy for the amount of kinetic energy that is induced by the perturbation of the generator electric angle in the form of a step change and whose dissipation is required for the system frequency to arrive at a new steady state (see Supplementary Note 8 for more information about frequency response and hertz-sec metric).

The randomness of generator inertia and damping coefficients represent technological variations that emulate both conventional and inverter-based renewable generation.

The randomness of loading condition resembles the loading variations that determine node voltages and the flow of power across the power lines in a power network and, therefore, represents the varying network condition states.

The results are summarized in Fig. 5.

In all of the power networks studied, 100% of the cases where power flow converged to a stable equilibrium point (meaning that load and generation matched while adhering to voltage and thermal line limits) resulted in a dynamically stable case. This observation is consistent for the complete range of varying network inertia and damping values examined and this confirms the feasibility of operating electric power networks on up to 100% renewable-based generation, explained as follows. Any power network relies on multiple forms of electricity generation and in a 100% renewable generation scenario, this mix will likely include solar, wind, and hydro power. While hydro power uses synchronous generators with fixed inertia and damping coefficients, almost all other forms of renewable sources rely on power electronics, including GFM and GFL. The GFM has the capability to independently offer the desired inertia and damping coefficients through digital fast responding controllers. The variable and uncertain nature of most renewable resources as constrained by meteorological conditions and time of the day necessitates rapid switches between the generation technologies in order to utilize the available resources and serve the load demand without interruptions[3–5]. Therefore, such networks will have to operate with varying levels of inertia[48,49] and damping. From a synchronization perspective, the constraining limitation is the requirement for a non-GFL generator to always be interconnected to the grid in order to satisfy the necessary stability condition. This condition can easily be satisfied by GFM IBRs or hydro power plants. Hence, the stable synchronized operation of a power grid with 100% renewable-based generation is feasible, from a small-signal stability perspective. We emphasize here that a stable case is a network whose response is bounded, and thus analytically stable in the sense of Lyapunov's first theorem and not in the sense of practice. In practice, the conformance to operational standards and procedures could introduce many non-linear elements such as under-frequency load shedding, frequency ride-through, etc.

In our results, lower scores for hertz-sec (meaning smaller frequency transients) are witnessed for low-inertia high-damping conditions; whereas the higher scores for hertz-sec (meaning larger frequency transients) are recorded for high-inertia low-damping conditions, as consistently observed in all 6 power grids considered. Moreover, the damping appears to be a more prevalent factor than the inertia as the variation of hertz-sec scores coincides with the variation of damping more closely than with the variation of inertia. These results speak to the relative importance of damping in network stability, as compared to the commonly studied inertial aspect, and further validate our

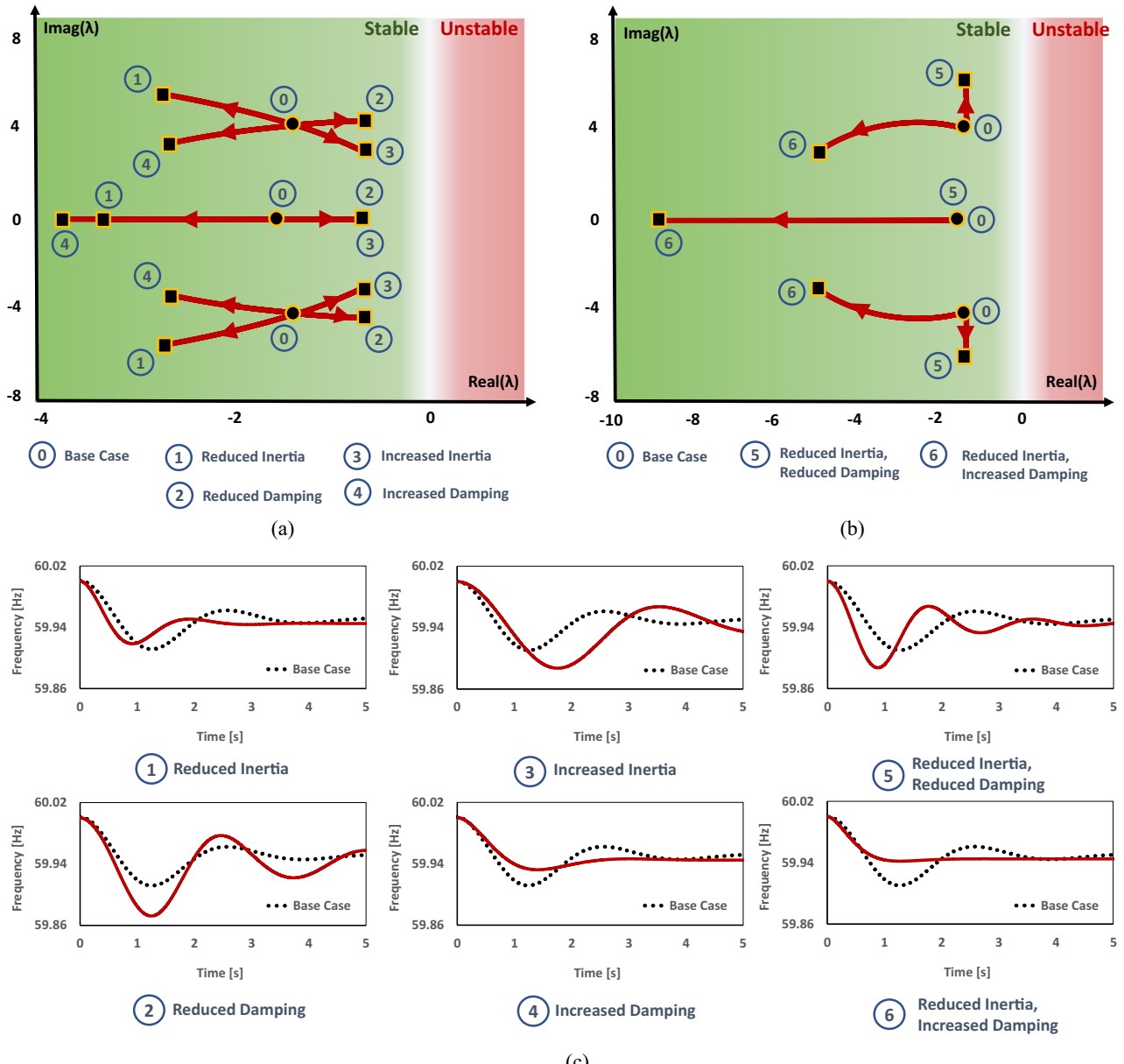

**Fig. 3 Loci of migration of eigenvalues in the complex plane and time-domain response as generator parameters are varied. a** Individual parameter changes, **b** concurrent parameter changes that replicated technological changes, **c** time-domain dynamic response. We analyzed the sensitivity of eigenvalues of a non-reference generator to parametric changes and the network coupling mode in the 3-generator network as a benchmark. For the base case, $M_i = (1.59, 0.80, 0.32)$ and $D_i = (1.53, 1.53, 0.92)$. For the network data see Supplementary Note 6. We considered six scenarios in addition to the base case. In each scenario, one or more parameters were changed by an order of 2. All plots shown include two internal modes that yield a pair of complex conjugate eigenvalues one coupling mode that is a pure real eigenvalue. In time-domain results, the higher the nadir frequency is, the more robust the frequency dynamic response (see Supplementary Note 8, for the characterization of frequency response trace). The results from individual parametric analysis here concluded that the reduction of inertia and the increase of damping moves the imaginary part of internal modes further away from the y-axis, resulting in an improved dynamic response, while the increase of inertia and reduction of damping has the opposite impact. For concurrent parametric analyses; scenario 5 represents the substitution of a synchronous generator with a grid-following (GFL) inverter and scenario 6 represents the replacement with a multi-loop droop grid-forming (GFM) inverter. These results exhibit the superior capability of the multi-loop droop GFM over GFL in improving the network dynamics. The GFM-VSM replacement is not presented here because it uses parameters equivalent to those of a synchronous generator and does not vary the result from the synchronous generator (SG) base case.

findings on the synchronization mechanism and the proposed synchronization enhancement strategy.

## Discussion
Power grids worldwide are changing significantly, transitioning from the currently dominant synchronous generator-based power plants to power electronics-based power plants in order to accommodate clean and renewable energy resources and achieve a decarbonized electric power sector. This transition is accelerating quickly as the cost of energy production using renewable power plants continues to decline[2].

In this work, we have derived the necessary condition for the stable synchronized operation of electric power networks,

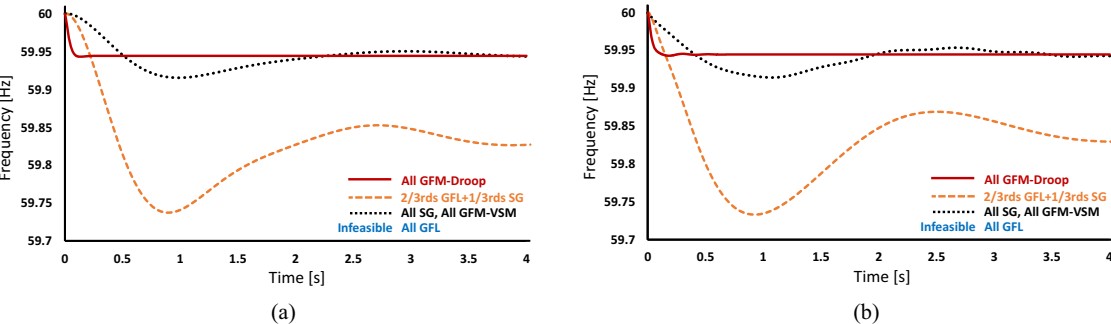

**Fig. 4 Validation of the analytical model using a high-order model implemented in an industry-grade power system planning software for electromagnetic transient (EMT) dynamic modeling parameters are varied. a** Results from generic second-order analytical models developed in this paper, **b** results from full-order models in Power Systems Computer Aided Design (PSCAD), available open-source[62]. The results here show the dynamic frequency response for a 3-generator, 9-bus network with different generation technologies. There are three main points of note. First, the frequency response for the three scenarios of "All Synchronous Generator (SG)", "All virtual machine grid-forming (GFM-VSM)", and "2/3rd grid-following (GFL)+1/3rd SG" are very similar. Second, operating a network with 100% GFL is infeasible because mathematically, the denominator of the Jacobian elements cannot be zero (100% GFL implies $M = 0$) and practically, there will be no source to construct the synchronization frequency for the GFLs to follow. Third, and perhaps the most significant point, when operating with 100% multi-loop droop grid-forming (GFM), the effective inertia value is reduced to near zero and the GFM frequency response can be seen effectively as first-order and therefore be less likely to experience severe frequency excursions. This corroborates with the observations reported in ref. [63]. The inertia and damping coefficients for these cases and the quantification of their dynamic response are presented in Supplementary Table II.

**Table 1 Dynamic performance of the cases considered when subjected to a small perturbation.**

| Case no. | Inertia coefficient | Damping coefficient | Natural frequency of oscillation ($\omega_n$) | Natural damping ratio ($\zeta$) | $\frac{\omega_n}{\omega_{n_0}}$ | $\frac{\zeta}{\zeta_0}$ |
|---|---|---|---|---|---|---|
| Base Case | $M_i$ | $D_i$ | 3.40 | 0.31 | 1 | 1 |
| Case 1 | $\frac{1}{2}M_i$ | $D_i$ | 4.76 | 0.43 | $1.40 \approx \sqrt{2}$ | $1.39 \approx \sqrt{2}$ |
| Case 2 | $M_i$ | $\frac{1}{2}D_i$ | 3.44 | 0.16 | $1.01 \approx 1$ | $0.52 = \frac{1}{2}$ |
| Case 3 | $2M_i$ | $D_i$ | 2.42 | 0.22 | $0.71 \approx \frac{1}{\sqrt{2}}$ | $0.71 \approx \frac{1}{\sqrt{2}}$ |
| Case 4 | $M_i$ | $2D_i$ | 3.28 | 0.59 | $0.96 \approx 1$ | $1.90 \approx 2$ |
| Case 5 | $\frac{1}{2}M_i$ | $\frac{1}{2}D_i$ | 4.85 | 0.22 | $1.42 \approx \sqrt{2}$ | $0.71 \approx \frac{\sqrt{2}}{2}$ |
| Case 6 | $\frac{1}{2}M_i$ | $2D_i$ | 4.46 | 0.79 | $1.31 \approx \sqrt{2}$ | $2.55 \approx 2\sqrt{2}$ |

Here, $\omega_0$ and $\zeta_0$ are the natural frequency of oscillation and the natural damping ratio in the base case, respectively (see "Methods" for definition). The coefficients for the base case is highlighted in the caption of Fig. 3. Values of $\frac{\omega_n}{\omega_{n_0}}$ and $\frac{\zeta}{\zeta_0}$ quantify how the natural frequency of oscillation and the natural damping ratio in each subsequent case changes relative to those of the base case. The results here conclude that the natural frequency of oscillations is independent of the generator damping coefficient, $D_i$, while the inverse square root of the inertia coefficient, $M_i$, determines the frequency of oscillations. They also indicate the natural damping ratio is a function of the generator damping coefficient $D_i$, with a direct relationship, and their inertia coefficient, $M_i$, is a square root relationship. The results presented in this table are notable as they underline the significance of the damping component in achieving a stable synchronized frequency response.

considering both conventional and renewable generation technologies, when subjected to small perturbations. We identified the adjustment of damping component of GFM-inverters as primary mechanism to enhance grid synchronization, particular during low-inertia operating conditions. While the common, contemporary dialog in research has tied network stability primarily to mechanical inertia, a physical characteristic of synchronous generators, here we have demonstrated the instead critical role of generator damping in achieving, and maintaining, a stable synchronized state of operation. The dynamics of emerging power grids will heavily depend upon the technology used for the interconnection of the renewable generation, whether GFL or GFM.

The findings here are important for research and development in decarbonized power grids and may set a scientific basis for the study of stability, dynamics, control, and operation of bulk power networks with 100% renewable-based generation. In particular, our theoretical and numerical results prove that the power networks with 100% renewable generation possess stable synchronization measures for dealing with small-signal perturbations, but their dynamic response may violate the necessary industrial control and operation standards; i.e., under frequency load-shedding (UFLS) scheme, the standards set for rate of change of

frequency (ROCOF)-based relays, and frequency response obligation set by the balancing authorities. We emphasize that this is not a fundamental limitation and this shortcoming can be addressed and managed by advanced coordinated control networks that may leverage the stability enhancement mechanisms we have established in order to modify the network behavior and provide a desired response[50]. Power networks have been successfully managed and operated by hierarchical networked control for more than half a century[51,52]. If the control apparatuses in power networks are poorly tuned or poorly structured, they can exhibit negative damping and be destabilizing[53]. As a result, the control networks and structure for conventional power networks, which have been designed around the dominant synchronous generator dynamics, are likely inadequate for the operation of a 100% renewable generation-based grid because of the fundamentally different underlying dynamics. While we have identified the stability improvements via adjustments to damping and inertial parameters, conventional power networks are equipped with control and protection apparatuses whose parameters are adjusted infrequently, if ever, if even possible (i.e., the mechanical inertia and damping of a synchronous generator are physical properties particular to a design). Furthermore, the majority of inverter-based resources integrated into the grid with the

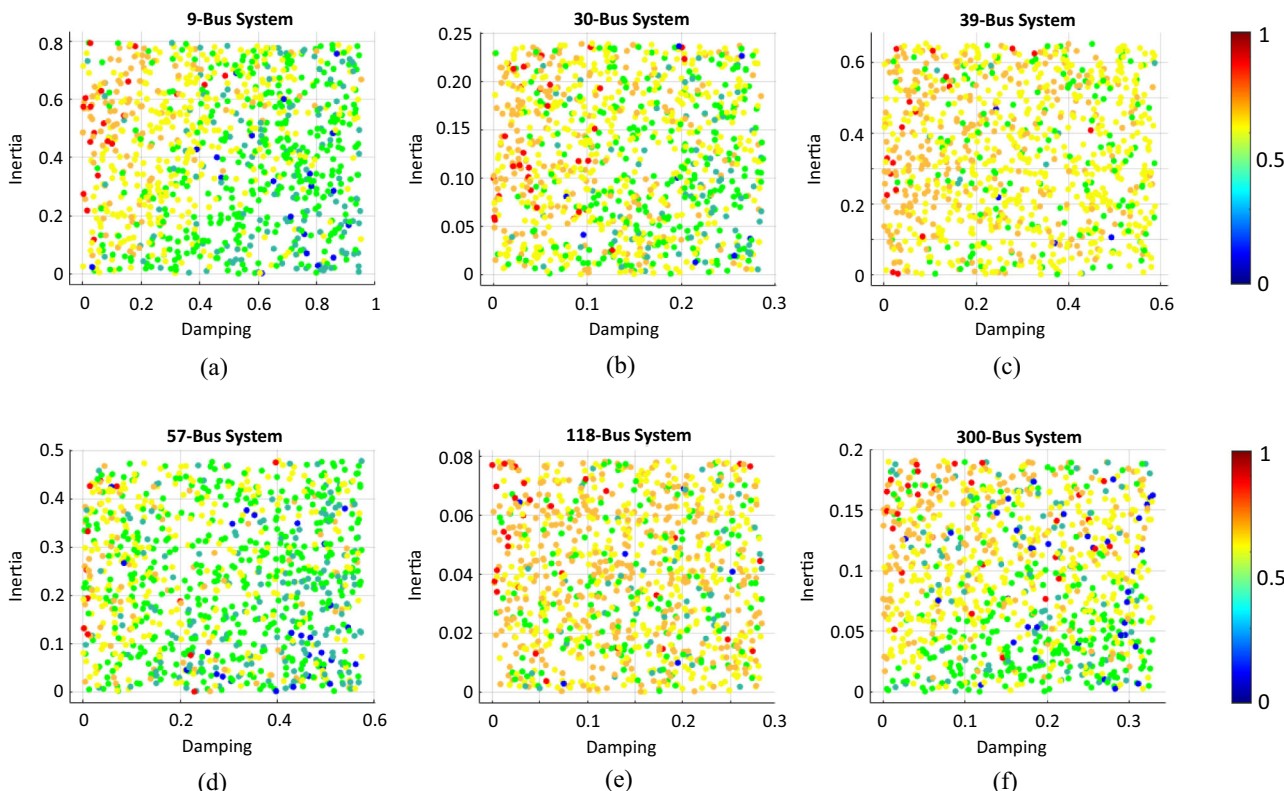

**Fig. 5 Normalized values of hertz-sec metric as a proxy for kinetic energy, as a function inertia and damping. a** 9-node, 3-generator test network, **b** 30-node, 6-generator test network, **c** 39-node, 10-generator test network, **d** 57-node, 7-generator test network, **e** 118-node, 54-generator test network, **f** 300-node, 69-generator test network. The results for 1000 random operating conditions on 6 different power network benchmarks are presented. The observed hertz-sec values are color-coded by the blue dots being the lower values indicating the best network dynamic response, and the red dots being the higher values reflecting the worst network dynamic response. The range of other colors should be interpreted accordingly as described by the color guide bar. The x- and y-axes represent the generators damping and inertia, respectively. In all six power networks, the trend of hertz-sec values is such that the improvement of frequency dynamics is evident, proportional to increased damping and reduced inertia (the best dynamic responses appear at the bottom of these plots and more towards right corner whereas the worst dynamic response appear in the top of these plots and more towards left corner). In addition, the impact of damping is more pronounced than that of inertia as the changes in score of hertz-sec are more directly proportional to the changes in damping value than to the changes in inertia value. The description of power networks and the source of their data is provided in Supplementary Note 9.

capability to adjust their parameters are often not obligated to provide frequency response support to the grid, because of both the technical capabilities (e.g., a lack of headroom reserve to respond) and the current real-time and ancillary services market structure (e.g., a lack of economic incentives or interconnection requirements).

Perhaps one of the most effective utilities of our work is to use it as the basis for the development of a control solution for the reliable, safe, and stable operation of future power networks. To this end, it is pivotal to reconsider the control and automation systems currently in place, both the structure and algorithms, and perhaps design and implement modern control systems that are designed and tuned in accordance with the dynamic behaviors and characteristics of power networks with high levels of inverter-based generation. In future power networks dominated by grid-forming inverters, new concepts such as adaptive protection that follows the grid inertia to adjusts its settings in real-time, and generators' available headroom reserve are factored into the determination of the droop value in real-time, making the grid a dynamically adaptive network and to do so, various control schemes can be utilized, especially highly distributed control systems. Such ideas are, of course, adaptive to the new structure of the grid with high shares of grid-supporting IBRs.

We recognize the solution established in this paper is an important part of a larger portfolio for the successful transition to

decarbonized power networks. While we addressed the dynamics associated with frequency synchronization at timescales of cycles to seconds, the decarbonization of power networks involves challenges on many timescales. On one end of the spectrum reside power balancing assurance and resource adequacy that fall under timescales of minutes all the way up to years. Planners study that problem in order to establish the capacity necessary to build years ahead, and operators look into this problem under unit commitment for day(s) ahead to ensure the generation capacity, reserve margin, and ramping capability are available to meet demand. On this timescale, high shares of renewable technologies call for more flexible resources, where many believe energy storage is the enabling technology. On the other end of the spectrum reside nonlinear transients that occur on timescales of milliseconds to cycles, as the system is continuously susceptible to high frequency switching events and potentially short-circuit faults. On this timescale, the high shares of renewable technologies require advanced prognosis, diagnosis, and isolation of impacted devices and clusters of the network. Across this spectrum, many more topics remain open questions such as power limiting, fault detection and diagnosis, power sharing, ancillary service, and market operation to hedge the risks associated with the variability and uncertainty of inverter-based resources.

We hope our findings bring forth a new perspective on emerging power networks and advance the grid planning and

optimization frameworks that take advantage of the unique functionalities, complexities, and responsiveness of power electronic devices.

## Methods

**Admittance of links**. The electrical power network interconnects power plants, substations, and load centers with electric power lines. The lines possess the electric characteristics of resistance, $R$, capacitance, $C$, and inductance, $L$. Given the timescale of our interest, one that is not on order of electromagnetic transients, one can assume the line parameters are constant. The total impedance of the line (link) that connects $i$th busbar (node) to $k$th busbar can be expressed by

$$z_{ik} = R + j2\pi fL - j(2\pi fC)^{-1}$$
$$= R + jX = |z_{ik}| \angle \vartheta_{ik} \ (\Omega) \tag{8}$$

with $j = \sqrt{-1}$ being the imaginary unit. The $X = X_L - X_C$ is the reactance in ohms, $X_L = 2\pi fL$ and $X_C = j(2\pi fC)^{-1}$ are inductance and capacitance in ohms, respectively. $|z|$ is the amplitude of the impedance (the Pythagorean sum in the complex plane) and $\angle \vartheta_{ik}$ is the respective angle in polar coordinates. The admittance of this link is the reciprocal of the impedance value and can be expressed by

$$y_{ik} = \frac{1}{z_{ik}} = \frac{1}{|z_{ik}| \angle \vartheta_{ik}}$$
$$= \left| \frac{1}{z_{ik}} \right| \angle \alpha_{ik} = |y_{ik}| \angle \alpha_{ik} \ (S) \tag{9}$$

with $|y_{ik}| = \left| \frac{1}{z_{ik}} \right|$ in siemens and $\alpha_{ik} = -\vartheta_{ik}$.

**Network admittance matrix**. The network admittance matrix, denoted by $Y$, is a square and symmetric matrix that represents the network topology and all associated links. It can be formed by allocating all off-diagonal elements connecting the $i$th to $k$th node, as the negative value of the corresponding admittance, $y_{ik}$, and the diagonal element for the $i$th node, $y_{ii}$ by the summation of all links connected to that node. For a network with $n$ nodes, it can be described as:

$$Y = \begin{bmatrix} \sum_{\xi=1}^{n} y_{1\xi} & -y_{12} & \cdots & -y_{1n} \\ -y_{21} & \sum_{\xi=1}^{n} y_{2\xi} & \cdots & -y_{2n} \\ \vdots & \vdots & \ddots & \vdots \\ -y_{n1} & -y_{n2} & \cdots & \sum_{\xi=1}^{n} y_{n\xi} \end{bmatrix} \tag{10}$$

**Network power flow**. Using Kirchhoff's circuit laws for voltage (KVL) and current (KCL), the equations that describe the steady-state equilibrium of an electric network are the power-flow equations described in (11).

$$p_{g_i} - p_{l_i} = v_i^2 \cdot G_{ii} + \sum_{k=1;k\neq i}^{n} v_i \cdot v_k \cdot [B_{ik} \cdot \sin(\theta_i - \theta_k) + G_{ik} \cdot \cos(\theta_i - \theta_k)] \tag{11}$$

where $v_i$ and $v_k$ are the voltage magnitudes at the $i$th and $k$th node and $\theta_i$ and $\theta_k$ their angles with $y_{ik}$ and $\alpha_{ik}$ the admittance value and angle of the branch that connects them, $p_{g_i}$ is the power injection into the $i$th node and $p_{l_i}$ is the power drainage at the same node. Solving this equation determines the generators power dispatch and voltage magnitudes and angles at all busbars (nodes). The power flow equations can be solved using iterative algorithms starting at an initial condition and reaching an equilibrium neighborhood by minimizing the error between two consecutive iterations[54]. Newton-Raphson and Gauss-Seidel are the most commonly used algorithms to solve power flow equations[54]. The solution of this analysis is an initial condition necessary for linearization and dynamic analysis. In this study, to obtain the numerical solution of power flow equations, we used the standard MATPOWER package[55].

**Kron reduction**. Kron reduction can be used to acquire a lower-dimensional electrically-equivalent network of a network when only boundary nodes, i.e., the nodes with a dynamic element directly interconnected, are kept[44]. This technique is extensively used in the analysis of dynamics of electric power networks with reduced dimensions of the network by eliminating nodes without a dynamic element directly attached[44,56–61]. Kron reduction produces an approximate admittance between boundary nodes—nodes with a dynamic element in this study—by approximating the admittance of an interconnecting link between nodes $i$ and $k$ with node $p$ being eliminated as[58]:

$$Y_{ik} = Y_{0_{ik}} - \frac{Y_{0_{ip}} Y_{0_{pk}}}{Y_{0_{pp}}}, \quad i \neq k, \ i,k = 1, \ldots, n \tag{12}$$

The resultant network is an equivalent power network in which all retained nodes are associated with dynamic generation. The eliminated nodes and loads are integrated through adjusted admittance values between the boundary nodes.

**Model linearization**. Linearization of nonlinear models involves an approximation of the system behavior around a particular equilibrium point by performing a

Taylor expansion and retaining only the first-order terms, which is valid to determine the system dynamics when subjected only to small perturbations. Suppose a nonlinear dynamical system expressed by

$$\dot{x}(t) = f(t, x(t), y(t), u(t))$$
$$0 = g(t, x(t), y(t), u(t)) \tag{13}$$

where dot notation indicates time derivative; $\dot{x} = \frac{d}{dt}x$, $t$ denotes time, $x$ is the vector of state variables, $y$ is the vector of algebraic variables, and $u$ is the vector of inputs. The linearized approximation of the system around $x^* = (x_0, y_0, u_0)$, assuming $u(t) = 0$, can be obtained by:

$$\Delta \dot{x}(t) = J \cdot \Delta x(t) \tag{14}$$

where $J = A_1 - \gamma_1 \gamma_2^{-1} A_2$ is the system Jacobian with $A_1 = \frac{\partial f}{\partial x}$, $\gamma_1 = \frac{\partial f}{\partial y}$, $A_2 = \frac{\partial g}{\partial x}$, and $\gamma_2 = \frac{\partial g}{\partial y}$.

**Second-order response characteristics**. The standard form of a mathematical description for a second-order dynamical system is

$$\frac{d^2 x}{dt^2} + 2\zeta \omega_n \frac{dx}{dt} + \omega_n^2 x = 0 \tag{15}$$

where $\omega_n$ is the natural frequency of oscillations which indicates the system natural mode, and $\zeta$ is the damping ratio.

**Hertz-sec metric**. To quantify the frequency response, we use the hertz-sec metric[47] which is a proxy for the changes of kinetic energy induced by the disturbance. This metric integrates the absolute value of frequency deviation over its transient period. The hertz-sec, $HS$, is defined as:

$$HS = \int_{t_0}^{t_s} |f_0 - f(t)| \cdot dt \tag{16}$$

where $f_0$ and $f(t)$ are the pre-disturbance frequency (a constant) and the post-disturbance frequency (a function of time), respectively, and $t_0$ and $t_s$ are the time the frequency departs the pre-disturbance steady-state value and the time it permanently reaches the settling steady-state value, respectively.

## Data availability

The raw data to generate Fig. 1 is restricted for public access due to regulation, and hence not shared. The data to generate Fig. 2 is embedded in its code. The data to generate Figs. 3 to 5 and Table 1 are deposited in Github and accessible via https://github.com/ahsajadi/sync. The PSCAD models used to produce results presented in Fig. 4 are available open-source on Github via https://github.com/NREL/PyPSCAD.

## Code availability

Codes for production of the results presented in Figs. 2 to 5 and Table 1 are available open-source on Github via https://github.com/ahsajadi/sync.

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

## Acknowledgements

We wish to thank C. Chen and Y. Liu with the University of Tennessee and Oak Ridge National Laboratory (ORNL) for providing us with the frequency measurement data of the European power network, R. Kolacinski with British Aerospace (BAE) Systems, J. D. Lara with the University of California Berkeley, B. Ravandi with the Northeastern University, R. Preece of the University of Manchester, and B. Wang with the National

Renewable Energy Laboratory (NREL) for their insightful discussions and comments. This work was authored in part by the National Renewable Energy Laboratory, operated by Alliance for Sustainable Energy, LLC, for the U.S. Department of Energy (DOE). The views expressed in the article do not necessarily represent the views of the DOE or the U.S. Government. The U.S. Government retains and the publisher, by accepting the article for publication, acknowledges that the U.S. Government retains a nonexclusive, paid-up, irrevocable, worldwide license to publish or reproduce the published form of this work, or allow others to do so, for U.S. This work was supported by the U.S. Department of Energy under Contract No. DE-AC36-08-GO28308 with the National Renewable Energy Laboratory.

## Author contributions

The conceptual approach and research design were performed by A.S. and R.W.K., with the majority of the analytical and numerical analyses carried out by A.S. B.M.H. supervised the project. All authors contributed to editing the manuscript.

## Competing interests

The authors declare no competing interests.
