## [Peer Review File · Nature Communications]

REVIEWER COMMENTS

Reviewer #1 (Remarks to the Author):

Feedback on the review

1. The noteworthy result of the study is the mathematically proved message that grid forming (referred to GFM) inverters representing the equation (2) are theoretically able to stabilize a nearly 100 % inverter based electrical grid. Because of the chosen preconditions for the investigation it is not clear whether the theoretical results meet the practical requirements. The critical preconditions are:

- a) No ohmic dissipation in the inverter and filter components (non-linear influence)
- b) The power set point value is only theoretically equal to the power export value (equation (3) or (8) in the supplement material). Therefore the given linear calculation only seems to be a first order approach.
- c) The definition of the damping in equation (7), (9) is not transparent enough (what are the meanings of the used different frequencies ω^* , ω , ω_i , $\Delta\omega_i$?). It is estimated that the damping is more or less a „droop damping“ (active) combined with a „machine damping“ (passive). A sign for that are the results shown in fig. 3c. There must be a coupling between inertia and damping function via the droop influence because the shown decreasing frequency by reducing damping (2) isn't convincing.

2. The study is a significant message for the grid stability in the case of replacement of real synchronous generators by virtual synchronous generators (VSG, GFM, VISMA etc.). The given literature in the study is not complete. A good overview of GFM types and their 15-years-old history is documented e.g. in ELSEVIER, Electric Power Systems Research by V. Mallema et al., 9/2021, 107516. It is recommended to add this literature source to the literature chapter of the study.

3. The study is based on an excellent approach of the very complex research questions concerning the grid stability in the future which are initiated by the transition from a synchronous generator-based grid operation to a new one which is nearly 100 % inverter-based with renewable generation. No conclusions, claims or additional evidences are necessary. But a more clear nomenclature, especially for the different frequencies mentioned in point 1c, is necessary, especially for distinction between rotor damping and droop frequency control. A control scheme in Laplace domain could be helpful.

4. In reality the rotor damping is completely decoupled from the droop control (there is a natural damping of the rotor in case of $\Delta\omega_i \neq 0$). The transfer function for rotor damping of the damper cage is not constant as assumed in equation (3) and (8). The characteristic is like a filter with an input signal coming from rotor oscillation ($\Delta\omega_i \neq 0$) with a time delay (parameter: complex synchronisation coefficient in real synchronous machines, Weh, ff. Elektrische Netzwerke und Maschinen in Matrizendarstellung, BI/108/108a, 1968). Before the study is published, it would be necessary to explain in detail why the inertia is influencing the steady state value of the frequency (fig. 3c). Especially in linear systems such a behaviour is strange.

5. The used linearized model is valid only for small signal operation which is not completely realistic.

From the engineer's point of view the theoretical study is very helpful, but more simulation results should be given to validate the analytics with a quantification of the derivation. This is necessary because up to now qualified experimental results haven't come in yet.

6. In conclusion it can be said that the given mathematical equations are convincing. However, the system details for modeling are still missing. They are important for the reproducibility. E.g. the mentioned (S6) 3-generator system (fig. 4) modelling is too short for reproducing the numeric validation of the used equations (5), (6), and (7). Therefore the understanding of the results shown in fig. 3 is not transparent enough.

Prof. Dr.-Ing. Hans-Peter Beck

Clausthal, 06.12.2021

Reviewer #2 (Remarks to the Author):

Manuscript NCOMMS-21-45155 analyzes the role of inertia and generator damping in the synchronization of the electric grid. The grid is currently dominated by conventional synchronous generators with a high inertia, but the future grid with a high penetration of variable renewable energy sources (RES), based on power electronics, will have a much reduced inertia. The reduction of the inertia of the system is, in principle, known to be detrimental for its stability. The authors address this problem studying the stability of the synchronized state in a simple model of coupled oscillators that takes into account the inherent synchronization dynamics only, considering the frequency response without primary or secondary frequency control. The main result of the manuscript is showing that reducing the inertia is actually beneficial for the robustness of the synchrony if it comes together with an increase of the generator damping.

The discussion of the role of the inertia and damping in the stability of a 100% renewable energy power grid is certainly interesting and timely. However, I'm not sure the results of this manuscript are significant enough to deserve publication in Nature Communications. The presented results are actually quite simple and not surprising. It is well known that, keeping the coupling strength constant, the lower the inertia the easier to synchronize two oscillators, as they will follow the applied force faster. Then, increasing the damping, of course ensures a faster convergence of the fixed point. This part, illustrated by Fig. 2, is then quite straightforward.

The issue with the inertia, in my opinion, does not have mainly to do with what discussed in the manuscript regarding synchronization. Inertia, what it does is increasing the response time of the system to demand changes, giving more time to the automatic control systems to react (with the same control capabilities, a demand-supply unbalance will create a smaller frequency deviation with higher inertia). Reduced inertia, at least during a transient period where RES have to coexist with conventional control methods, might still be a problem. The interaction of the response time of the system with reduced inertia with the response time of the (conventional) control methods is what makes frequency control a difficult task. With this regard, the authors say that increased damping is the solution, and I agree to some extent. Damping is, essentially, a very fast primary control. While this can be achieved by power electronics easily, it is not the case for conventional generators, and in periods of low RES production, if conventional backup generators must be brought online, the control of the system may be in danger. Also, if a demand-supply unbalance is kept for a relatively long time, huge amounts of storage will be needed, which may not be cost-competitive, at least in the short run, with conventional control strategies. All these issues can not be addressed in the framework of the studied model, but at least it should be discussed in the manuscript.

Minor remarks:

1. Table I: The label t_s does not correspond with what used in the caption for the peak time (t_p).
2. Page 10, below Table I: "has been widely been characterized"
3. Fig. 3. The position of the labels (a), (b), (c) is a bit confusing. Also, in the caption, is the sentence "moves the imaginary part of internal modes farther away from the y-axis" correct?. Finally, the caption mention the case "increase of inertia and reduction of damping" but the label 5 of the figure says "reduced inertia, reduced damping".
4. In the discussion section, what do the author exactly mean with the sentence "but their dynamic response may violate the necessary industrial control and operation standards"? Please elaborate.

Reviewer #3 (Remarks to the Author):

The article "Inherent Synchronization in Electric Power Networks with near 100% Inverter-Based Renewable Generation" discusses conditions of synchronization of power grids and claims that such an operation is possible with 100% inverter-based renewable generation. The authors investigate how a power grid, in particular, one with heterogeneous parameters and potentially a high share of inverters can synchronize and thereby run stably without any added control action.

The topic is of great interest for an interdisciplinary audience and the quality of the figures and text is high. I feel some claims made by the authors are too strong based on the results presented and the presentation can be improved.

Overall, I recommend publication after revision.

Specific comments and questions:

1. The title suggests a different focus than the abstract and then the actual results: Both title and abstract should reflect that the focus is on synchronization criteria and stability considerations of networks with heterogenous parameters and potentially grids that are based 100% on inverter-based generators.
2. I am not convinced that the last paragraph in the introduction is necessary. Instead, I would have expected some overview of the paper and its structure.
3. What data was used to create Fig1? Did the authors themselves have PMUs in the CE grid?
4. On page 5 the authors implicitly utilize the DC approximation as they state the transmission line dynamics is governed by the angle difference (instead of the sine of the angle difference). Not even in the methods do the authors mention the term "DC approximation".
5. Follow-up of point 1: Only around pages 5-6 the authors indicate that heterogeneous parameters are critical. Why didn't they mention this point already in the title, abstract, introduction?
6. When discussing equation (2): The authors include a "damping term" but explicitly exclude automatic control. It seems primary/droop control is used and it should be mentioned in the discussion of the results that increasing damping (or in this case droop control) is stabilizing for the system. Since droop control is adding power to the system this result is expected.
7. "Given our timescale of interest in this study, it is assumed that the load is constant." What exactly is the time scale of interest here?
8. Equation (4) is confusing: Firstly, the difference between ω_i and ω_n is not clear. Secondly, the dimension of all entries should be indicated more clearly. In particular,

the "0" entries to node have the same dimension but can be $1 \times (n-1)$ or $(n-1) \times (n-1)$ or $(n-1) \times 1$. The -1 matrix also seems to be $(n-1) \times 1$, based on Supplementary Equation (25). Please check the notation and make it easier to read and process.

9. How can equation (5) as a third-order polynomial yield $2n-1$ solutions for lambda?

10. The results in Table I seem suspicious: The authors claim that with only inverters and thereby much smaller damping and inertia the nadir and return times stay essentially identical? Also, the scenarios are not really introduced or discussed in the text. Potentially this table could even be moved to the Supplements as it does not seem to contain important results for the main text.

11. The main text should at least mention the nature of the "small perturbation" is this a perturbation in power, angle, frequency or some system parameter? Does it persist or is it only active at one time instance? Checking the methods and supplements, I still do not know how this "small perturbation" looks like.

12. The authors claim "this confirms the feasibility of operating electric power networks on 100% renewable based generation". However, they have only shown that power grids can synchronize using inverter-based technology. While the authors claim that on the time scale they are interested in (and do not specify) the load will not change, renewable generation can easily change on the time scale of few seconds, minutes etc. None of the results presented here tackle the balancing problem of renewable generation. Hence, I urge the authors to make their claim more moderate and specific to fit what they have actually shown.

13. During the discussion the authors write "As a result, the control networks and structure for conventional power networks, which have been designed for a synchronous generator-dominated network to manage these machine unique dynamics, are likely inadequate for the operation of a 100% renewable generation-based grid because of the different underlying dynamics." But how is this statement backed up by the results of this study? This seems like pure speculation.

14. Connected to the previous point: How does the current paper act as a "basis for the development of a control solution"? By giving the case without control so that further research can add control terms?

15. There are a couple of typos/unclear expressions, e.g. "to damp out the transient forced induced in spring", "GFM as applied in parallel on power systems an emerging and promising technology." Especially in the captions, sentences tend to be very long and complicated. Please check and see whether they can be simplified.

16. Finally, I ask the authors to include a code and data availability statement and publish their code for transparency and reproducibility reasons in a publicly accessible way (e.g. github, OSF, zenodo...)

Response to Reviewer #1

Reviewer #1, Comment 1: The noteworthy result of the study is the mathematically proved message that grid forming (referred to GFM) inverters representing the equation (2) are theoretically able to stabilize a nearly 100 % inverter based electrical grid. Because of the chosen preconditions for the investigation it is not clear whether the theoretical results meet the practical requirements. The critical preconditions are:

- a) No ohmic dissipation in the inverter and filter components (non-linear influence)
- b) The power set point value is only theoretically equal to the power export value (equation (3) or (8) in the supplement material). Therefore the given linear calculation only seems to be a first order approach.

Author's Response: We appreciate your feedback and observations. The models presented in this paper are for system-level studies and therefore use aggregated device models. The aggregated models take into account those dynamics within devices, including the filters that contribute to the frequency response (See Supplement Material, Page 5, Figure 5). Nonetheless, the aggregated model control objective is to bring the error between the actual output power vs. output power set point down to zero. Additionally, the ohmic losses, as related to the dissipation of power within the inverter, and do not alter the frequency dynamics at the system-level.

Reviewer #1, Comment 2: c) The definition of the damping in equation (7), (9) is not transparent enough (what are the meanings of the used different frequencies ω^* , ω , ω_i , $\Delta\omega_i$?). It is estimated that the damping is more or less a „droop damping“ (active) combined with a „machine damping“ (passive). A sign for that are the results shown in fig. 3c. There must be a coupling between inertia and damping function via the droop influence because the shown decreasing frequency by reducing damping (2) isn't convincing.

The frequency related variables in Eq. (7) and Eq. (9) of the supplemental material are described as follows:

- ω is the generic notation for frequency,
- ω^* denotes the frequency set-point,
- ω_g denotes the measured frequency,
- ω_i denotes i th generator's frequency as a nonlinear variable, and
- $\Delta\omega_i$ denotes i th generator's frequency as a linear variable (Δ is the linear difference operator)

We have updated the text of the Main manuscript and the Supplemental material so that these variables are better explained and distinguished.

In Eq. (7) and Eq. (9) of the Supplemental Material, we define the damping as the aggregation of elements that contribute to the dissipation of induced frequency response-related oscillations. Analytically, we derived the roots of the

characteristic equation for the closed loop feedback system that describes the changes of frequency to changes of power output ($\frac{\Delta\omega}{\Delta P_L}$) as

$$s_{1,2} = \frac{1}{2} \left(- \left(\frac{1}{T_{TG}} + \frac{D_m}{R \cdot M_m} \right) \pm \sqrt{\left(\frac{1}{T_{TG}} + \frac{D_m}{R \cdot M_m} \right)^2 - 4 \left(\frac{D_m}{M_m \cdot T_{TG}} + \frac{K_{TG}}{R \cdot M_m \cdot T_{TG}} \right)} \right)$$

where

D_m is the mechanical damping torque by the damper windings,

M_m is the rotor mechanical inertia

R is the droop gain

T_{TG} is the governor and turbine response time constant

K_{TG} is the governor and turbine response gain constant

Looking at the roots, ($s_{1,2}$), the damping component is mainly determined by the real part of these roots, $-\left(\frac{1}{T_{TG}} + \frac{D_m}{R \cdot M_m}\right)$. According to this relationship, for a synchronous generator, the components that contribute to damping are the:

- (1) mechanical damping provided by the damper winding,
- (2) product of droop gain and mechanical inertial momentum, and
- (3) the response time constant of governor and turbine system.

The mathematical description of the frequency response of a synchronous generator concludes the following: the larger the mechanical dampers, (D_m), and/or the smaller the product of the droop gain and mechanical inertia, ($R \cdot M_m$), and/or the smaller governor and turbine response time, (T_{TG}), (indicating a faster response time), the larger the magnitude of the negative real part of the imaginary number, which indicates a higher damping of the natural modes.

The results from the sensitivity analysis shown in Fig. 3 of the Main manuscript are purposed to indicate how variation of individual aggregated parameters can result in the migration of eigenvalues on the complex plane, and subsequently, a different time-domain response.

We have expanded Section S2 of the Supplemental material and explained this relationship between the damping and inertia through droop.

Reviewer #1, Comment 3: The study is a significant message for the grid stability in the case of replacement of real synchronous generators by virtual synchronous generators (VSG, GFM, VISMA etc.). The given literature in the study is not complete. A good overview of GFM types and their 15-years-old history is documented e.g. in ELSEVIER, Electric Power Systems Research by V. Mallema et al., 9/2021, 107516. It is recommended to add this literature source to the literature chapter of the study.

Author's Response: We are very grateful for the reviewer sharing this article with us. Accordingly, we have expanded upon the literature review. In the revised

main body of manuscript, more details of GFM is included and in the Supplemental material, section S2, power electronics subsection, we have highlighted various GFM technologies. Additionally, the recommended paper (V. Mallemaci et al., 9/2021, 107516, Electric Power Systems Research, Elsevier) is cited and referred to as a source of further detailed information.

Reviewer #1, Comment 4: The study is based on an excellent approach of the very complex research questions concerning the grid stability in the future which are initiated by the transition from a synchronous generator-based grid operation to a new one which is nearly 100 % inverter-based with renewable generation. No conclusions, claims or additional evidences are necessary. But a more clear nomenclature, especially for the different frequencies mentioned in point 1c, is necessary, especially for distinction between rotor damping and droop frequency control. A control scheme in Laplace domain could be helpful.

Author’s Response: We appreciate your recognition of the merits our paper has to offer. And thank you for your suggestion about the nomenclature. Accordingly, we have added a nomenclature for the different frequencies mentioned in point 1c. We hope the esteemed reviewer finds this addition satisfactory.

In the revised Supplement Material, we also have added a **Laplace block diagram** of the control processes that forms the aggregated model we used for synchronous generator and the multi-loop grid-forming technology, as shown below, in Fig. R 1.

(i) Synchronous generator (ii) Multi-loop grid-forming inverter

Fig. R1. Laplace block diagram of the aggregated model for frequency response

We verified and validated these reduced-order models for studying frequency response using the full-order models in PSCAD, as shown in Fig. R2. The results show that these reduced order models are quite accurate in producing the SG and GFM frequency response.

— Full-order Model
 - - Reduced-order Model

(a) SG

— Full-order Model
 - - Reduced-order Model

(b) GFM

Fig. R2. Validation of the analytical model using full-order model implemented in an industry-grade power system planning software for electromagnetic transient modelling (EMT) dynamic modelling, per generation technology

Given the transfer function of both the SG and the GFM reduced-order models (those shown in the above Laplace block diagram) yield a second-order transfer function, next, we introduced a generic second-order model in the form of

$$\begin{aligned} \dot{x}_1 &= x_2 \\ \dot{x}_2 &= M^{-1}(P^* - P_e - Dx_2) \end{aligned}$$

that we used to represent both SG and GFM in which M and D , are the inertia and damping components that represent all contributing elements (the values of M and D vary for SG vs. GFM technology). We validated the generic models by comparing their resultant frequency response with those from the reduced order models presented in the Laplace diagram with explicit elements included (see Fig. R3 below).

- - Reduced-order Model
 — Generic Second-order Model

(a) SG

- - Reduced-order Model
 — Generic Second-order Model

(b) GFM

Fig. R3. Validation of the analytical generic second-order models by comparing the resultant frequency response with those from the reduced-order model (previously validated using the full-order model implemented PSCAD)

Reviewer #1, Comment 5: In reality the rotor damping is completely decoupled from the droop control (there is a natural damping of the rotor in case of $\Delta\omega \neq 0$). The transfer function for rotor damping of the damper cage is not constant as assumed in equation (3) and (8). The characteristic is like a filter with an input signal coming from rotor oscillation ($\Delta\omega \neq 0$) with a time delay (parameter: complex synchronisation coefficient in real synchronous machines, Weh, ff. Elektrische Netzwerke und Maschinen in Matrizendarstellung, BI/108/108a, 1968).

Author's Response: The esteemed reviewer highlights a very important point here. The droop damping is a constant, but the damping torque in the synchronous generator provided by the damper windings can vary slightly with the change of rotor angle. However, the damping power produced by the damping torque can be analyzed as a function of speed deviation, as shown in Figure R2. When $\Delta\omega = 0$, both d and q factors are equal to zero and the overall value of damping loss increases proportionally to the speed deviation for small speed deviations.

Since we study small-signal stability, referring to small perturbations, we are interested in the neighborhood close to $\Delta\omega = 0$ where the difference between d, q, and the average value is insignificant (shown in the green circle in Fig. R4).

To summarize the abovementioned points, for network studies it is the convention to assume the $D_m(\delta) \approx D_m$ as a constant value, a precedent set by J. Machowski et al at [R1] and P. Sauer et al at [R2] (two main reference textbooks of the field of power network dynamics), for two reasons. (1) the variations of damping power are linear within the regions near the origin and (2) the small, bounded variations of D_m with changes of δ simplify the problem,

Fig. R4. Average value of the damping power as a function of speed deviation, from [R1]

[R1] Machowski, J., J. W. Bialek, and J. R. Bumby. "Power System Dynamics. Stability and Control." (2008).

[R2] Sauer, P. W., and M. A. Pai. "Power System Dynamics and Stability, Champaign, IL." (2006).

In the Supplemental material, Section S2, we have highlighted this fact and now it reads as

"It should be noted that D_m is a function of rotor angle (δ), $D_m = f(\delta)$, though for small deviations from the synchronous speed, its variation is minimal and can be assumed a constant, $D_m(\delta) \approx D_m$ ".

Reviewer #1, Comment 5: Before the study is published, it would be necessary to explain in detail why the inertia is influencing the steady state value of the frequency (fig. 3c). Especially in linear systems such a behaviour is strange.

Author's Response: Thank you for your critical observation. The steady-state value of frequency response does not change with the changes of inertia. We updated the results shown in Fig. 3c of the Main manuscript, in which the steady-state value of frequency response is the same for all parametric variations.

Please note that the updated results in the revised manuscript differ from those included in the initial submission, and the difference between the results stem from using the new values for dynamic variables after validation of the model using the industry-grade EMT simulations.

Reviewer #1, Comment 6: The used linearized model is valid only for small signal operation which is not completely realistic.

From the engineer's point of view the theoretical study is very helpful, but more simulation results should be given to validate the analytics with a quantification of the derivation. This is necessary because up to now qualified experimental results haven't come in yet.

Author's Response: We agree with the esteemed reviewer on this point for further validation of the findings of this paper and the need for experimental results. For power grid analyses at the system level, it is always difficult to acquire experimental data because the power grids are intertwined with security and financial systems and, hence, there exist extensive restrictions around conducting experiments on the system itself. Because of this, the system planners and operators all around the world mainly rely on very granular modelling using industry-grade software packages. And we find it reasonable to follow the same procedure and provide further industry-grade simulations for validation purposes.

For this analysis, here we use the power system computer aided design (PSCAD) software package, which is an electromagnetic transient (EMT) program and is fully capable of capturing the fast dynamics of inverters and models all three phases with suitable time resolution (in the microsecond range) to capture transients.

We have utilized a set of industry-grade models in PSCAD format that were made available to the public as open-source for verification and validation of our analytical model proposed in this paper. These models are developed by the National Renewable Energy Laboratory (NREL) and the University of Colorado Boulder in the United States, available at <https://github.com/NREL/PyPSCAD>. The description of the analytics behind this model is available at <https://www.nrel.gov/docs/fy21osti/78402.pdf>

In the revised Main manuscript, we have provided additional simulation results, shown in Fig. 4 of the Main document, and have been able to successfully validate and verify our analytical model presented in this paper using the PSCAD industry-grade models of the IEEE 9-bus system, as shown below in Figs. R5 and R6. It can be seen the frequency response traces virtually match.

(i) Results from the generic second-order analytical models developed in this paper

(ii) Results from full-order PSCAD models - industry-grade software

Fig. R5. Validation of the analytical model using full-order models implemented in an industry-grade power system planning software for electromagnetic transient modelling (EMT) dynamic modelling

(a) All SG

(b) 2/3rd GFL+1/3rd SG

(c) All GFM

Fig. R6. Validation of the analytical model using full-order models implemented in an industry-grade power system planning software for electromagnetic transient modelling (EMT) dynamic modelling, per generation technology

Reviewer #1, Comment 7: In conclusion it can be said that the given mathematical equations are convincing. However, the system details for modeling are still missing. They are important for the reproducibility. E.g. the mentioned (S6) 3-generator system (fig. 4) modelling is too short for reproducing the numeric validation of the used equations (5), (6), and (7). Therefore the understanding of the results shown in fig. 3 is not transparent enough.

Author’s Response: We indeed concur with the esteemed reviewer and are supportive of research reproducibility. The analytical modelling of the 3-generator system has been included in detail in **Supplemental Material, S6**.

In the revised manuscript, in **Supplemental Material, S6**, we have added the dynamic data and the transmission line and transformer data used for production of Fig. 3 of the Main document. By simply plugging in the data into the mathematical model after applying Kron reduction, we trust that interested readers should be able to reproduce the results presented in this paper, such as Fig. 3 and Fig. 4.

Fig. R7. One-line diagram of the 3-generator, 9-bus system, and the dynamic and network data

TABLE II: Parameters for the cases presented in Fig. 4 of the Main Manuscript

Case	Inertia Coefficient	Damping Coefficient	Nadir	ROCOF
All SG/GFM-VSM	(1.9099, 0.9549, 0.3820)	(1.8335, 1.8335, 1.1001)	59.91	0.276
2/3rd GFL+1/3rd SG	(1.5915, 0.7958, 0.3183)	(1.5279, 1.5279, 0.9167)	59.73	0.376
All GFM-Droop	(0.0080, 0.0040, 0.0016)	(0.3361, 0.3361, 0.2017)	59.94	0.720

To reproduce the results shown in Figure 3, the data from All SG/GFM-VSM should be plugged into the model. The network data are available in Fig 10 of the Supplemental material and Figure R7 of this response letter.

For further transparency and reproducibility, we have pledged to the Associate Editor that we will make all the datasets that are used to produce the results presented in this paper available to the public open source, if we are to receive a recommendation to publish in this journal and ahead of the final production of the paper. However, at this stage in the review process, to prevent the authors' identities from being revealed and comply with the double blind-review process policies, we are not yet able to do so.

Response to Reviewer #2

Reviewer #2, Comment 1: Manuscript NCOMMS-21-45155 analyzes the role of inertia and generator damping in the synchronization of the electric grid. The grid is currently dominated by conventional synchronous generators with a high inertia, but the future grid with a high penetration of variable renewable energy sources (RES), based on power electronics, will have a much reduced inertia. The reduction of the inertia of the system is, in principle, known to be detrimental for its stability. The authors address this problem studying the stability of the synchronized state in a simple model of coupled oscillators that takes into account the inherent synchronization dynamics only, considering the frequency response without primary or secondary frequency control. The main result of the manuscript is showing that reducing the inertia is actually beneficial for the robustness of the synchrony if it comes together with an increase of the generator damping.

Author's Response: We thank the reviewer for the recognition of merits of our paper and research.

Reviewer #2, Comment 2: The discussion of the role of the inertia and damping in the stability of a 100% renewable energy power grid is certainly interesting and timely. However, I'm not sure the results of this manuscript are significant enough to deserve publication in *Nature Communications*. The presented results are actually quite simple and not surprising. It is well known that, keeping the coupling strength constant, the lower the inertia the easier to synchronize two oscillators, as they will follow the applied force faster. Then, increasing the damping, of course ensures a faster convergence of the fixed point. This part, illustrated by Fig. 2, is then quite straightforward.

Author's Response: We sincerely appreciate the esteemed reviewer for agreeing with us on the criticality and timeliness of our research and this very important topic the paper addresses. We are also grateful for raising to our attention the need for better clarification of the novelty and significant impact of this work, as we might have not explained these details in our original submission. We are also very grateful for the opportunity to further work on this manuscript for its betterment. Accordingly, we have revised the paper and in the revised manuscript, we have done our best to directly address the concerns raised by the esteemed review and, therefore, we hope that the revised manuscript better explains the novelty, the broad impact, and application of our proposed solution.

We kindly ask the esteemed reviewer to reconsider this paper, taking into account the following points, and afford us the opportunity to share our findings with the readers and science community through publication in *Nature Communications*.

- 1) We agree with the esteemed reviewer that the clarity of presentation of results and the elegance of our solution is a core strength which allows the readers to well understand this problem and solution and also to extrapolate further ideas in this direction and continue to advance the field. Even though it may be deemed as simple (yet novel and precise), we should not forget it is true only to the experts and frontiers of this field.

- 2) We also agree with the esteemed reviewer that our results are intuitive to the scholars who are versed in complex networks and system dynamics in fact provides corroboration to the accuracy of our results and findings.
- 3) Furthermore, we agree with the esteemed reviewer that our approach is rooted in classical mechanics yet the solution we have established in this paper is addressing a critical problem in electric power networks. Therefore, this naturally paves the path for our contribution to be recognized as cross-and multi-disciplinary contribution by advancing the two fields of (i) complex networks by establishing the sufficient conditions for synchronization in networks with heterogenous parameters and (ii) electric power systems by establishing the critical role of damping in achieving a stable frequency response as opposed to what the existing body of literature refers to as the need for maintenance of inertia. The current literature is centered on a field of “low-inertia power systems” has been born, which yields the necessity for further research to find solution to cope with “reduced inertia” as a seemingly problematic issue. Our paper fundamentally challenges the hypothesis that low inertia is the foremost concern and brings forth a new direction of research by highlight the importance of damping.
- 4) Our results and findings have broader impact than the specific application addressed in this paper and the proposed solution applies to all dynamical networks with heterogeneous subsystem/subnetwork characteristics.

On one hand, our paper for the first time advances the conditions of stability by consideration of inherently heterogenous subsystem/subnetwork characteristics that previously had not been accomplished; the state of the literature currently resides at limitations of a homogenous damping factor [R3] (itself published in a Nature journal) and our solution has advanced this closer towards the reality seen every day in functioning power systems. This is an important next step given the wide variety of generator types (and thus damping factors) in both traditional and high-inverter power systems. We also use the mathematical machinery to form a convenient closed form solution for the Jacobian of the system such that the coupling mode and oscillatory modes are naturally decomposed. This allows the unique demonstration of the mechanism for the enhancement of stability by the adjustment of damping and inertia in complex networks of this kind.

On the other hand, our paper, for the first time, leveraging the formalism developed in this paper, presents an analytical model that allow the quantification of the criticality of damping in achieving a stable frequency synchronization. We recognize that recently the notion of damping-like support that grid-forming inverters can provide has been holistically alluded to in the literature [R4-R6], but thus far nowhere in the literature has a direct relationship been derived. And that is what our paper offers by deriving a direct mathematized relationship between not only inertia, but also damping, components in the grid and the frequency stability and synchronization. This allows us to not only analytically establish the feasibility of operation of power system with extremely low levels of inertia, but also, for the first time to the best knowledge of the authors, draws the attention to

the agile damping that power electronics offer and the crucial role it can play in future inverter-dominated power systems.

We suggest that our findings here have a broader impact across various disciplines involving dynamical systems and complex networks, particularly those that resemble multi-body problem or interconnected harmonic oscillators with the heterogenous subsystem/subnetwork characteristics, such as a network of connected vehicles, smart cities, financial networks, traffic networks and connected vehicles, networks of diseases and infections, etc.

[R3] A. Motter, et al. “Spontaneous synchrony in power-grid networks.” *Nature Physics* 9.3 (2013): 191-197.

[R4] R. H. Lasseter, Z. Chen, and D. Pattabiraman, “Grid-Forming Inverters: A Critical Asset for the Power Grid,” *IEEE Journal of Emerging and Selected Topics in Power Electronics*, vol. 8, no. 2, pp. 925–935, Jun. 2020, doi: 10.1109/JESTPE.2019.2959271.

[R5] U. Markovic, O. Stanojev, E. Vrettos, P. Aristidou, and G. Hug, “Understanding Stability of Low-Inertia Systems,” Feb. 2019. [Online]. Available: <https://doi.org/10.31224/osf.io/jwzrq>

[R6] D. Pattabiraman, R. H. Lasseter., and T. M. Jahns, “Comparison of Grid Following and Grid Forming Control for a High Inverter Penetration Power System,” in 2018 IEEE Power Energy Society General Meeting (PESGM), Aug. 2018, pp. 1–5. doi: 10.1109/PESGM.2018.8586162

Reviewer #2, Comment 3: The issue with the inertia, in my opinion, does not have mainly to do with what discussed in the manuscript regarding synchronization. Inertia, what it does is increasing the response time of the system to demand changes, giving more time to the automatic control systems to react (with the same control capabilities, a demand-supply unbalance will create a smaller frequency deviation with higher inertia). Reduced inertia, at least during a transient period where RES have to coexist with conventional control methods, might still be a problem. The interaction of the response time of the system with reduced inertia with the response time of the (conventional) control methods is what makes frequency control a difficult task. With this regard, the authors say that increased damping is the solution, and I agree to some extend.

Author’s Response: Indeed, we concur! We both share the view that additional damping is the solution to future grids with 100% renewables. This has been an overlooked portion of the research on “low-inertia” power systems, which we hope that this paper can help address.

Beyond the correct understanding that inertia provides a time-scale bridge between governor response and power imbalance, the retarding effect also prevents substantial deviations of the relative angles between generator buses. The maintenance of relatively small angle deviations is critical to synchronization

as the non-linear relationship created by the Sine of the angle deviations yields critical boundaries that cannot be breached.

Reviewer #2, Comment 4: Damping is, essentially, a very fast primary control. While this can be achieved by power electronics easily, it is not the case for conventional generators, and in periods of low RES production, if conventional backup generators must be brought online, the control of the system may be in danger. Also, if a demand-supply unbalance is kept for a relatively long time, huge amounts of storage will be needed, which may not be cost-competitive, at least in the short run, with conventional control strategies. All these issues can not be addressed in the framework of the studied model, but at least it should be discussed in the manuscript.

Author’s Response: We recognize the operational challenges that the esteemed reviewer raises to fall under the “energy imbalance” problem, which in power system planning is commonly called the resource adequacy problem, to ensure that MW generation (both to dispatch and for reserve) matches the forecasted consumption and in real-time market operation is known as the resource sufficiency problem. In this form it ensures that capacity and ramping capability among balancing authorities within a footprint exist. Of course, all these issues are part of a larger decarbonization portfolio that cannot be addressed in one paper.

In this work we do not try to address “the balancing problem” associated with renewable generation, only “the inverter problem”, as laid out by P. Denholm et al [R1]. The timescale of interest in the present paper is cycles to seconds, establishing feasibility and necessary conditions for the synchronized operation of an interconnected power network encompassing various synchronous generators, grid-following inverters, and grid-forming inverters.

Overall, power systems have temporally separated sets of decision-making that required multi-timescale measurement and operations. Figure R8 summarizes the multi-temporal dynamics of power system and highlights where in this spectrum our analysis falls.

Figure R8. Timescales in electric grid dynamics and operation, borrowed from [R2]

We updated the manuscript addressing this important topic and how our findings are only a piece of a larger portfolio. In the Discussion section, Page 15 of the main manuscript, we have added the following section.

“We recognize the solution established in this paper is an important part of a larger portfolio for the successful transition to decarbonized power networks. While we addressed the dynamics associated with frequency synchronization at timescales of cycles to seconds, the decarbonization of power networks involves challenges of varying timescale. On one end of the spectrum reside power balancing assurance and resources adequacy that fall under timescales of minutes and even years, as planners study that problem under resource adequacy in order to establish the capacity necessary to build years ahead, and the operators look into this problem under unit commitment for day(s) ahead to ensure the generation capacity, reserve margin, and ramping capability are available to meet the demand. On this timescale, high shares of renewable technologies call for more flexible resources, where many believe energy storage is the enabling technology. On the other end of the spectrum reside nonlinear transients that occur on timescales of milliseconds to cycles, as the system is continuously susceptible to short-circuit faults and high frequency switching events. On this timescale, the high shares of renewable technologies require advanced prognosis, diagnosis, and isolation of impacted devices and clusters of the network. Across this spectrum, many more topics remain open questions such as power limiting, fault detection and diagnosis, power sharing, ancillary service, and market operation to hedge the risks associated with the variability and uncertainty of inverter-based resources.”

[R1] P. Denholm et al, “The challenges of achieving a 100% renewable electricity system in the United States,” *Joule*, Volume 5, Issue 6, 2021, Pages 1331-1352,

[R2] A. Von Meier, “Integration of renewable generation in California: Coordination challenges in time and space” *Proceeding of 11th international conference on electrical power quality and utilisation* (pp. 1-6). IEEE, October 2011

Reviewer #2, Comment 6: Table I: The label t_s does not correspond with what used in the caption for the peak time (t_p).

Author’s Response: Upon the recommendation from the esteemed Reviewer 3, replaced Table I of the Main manuscript with a new table that presents more concise and important information about synchronized frequency oscillations and its constituting factors and moved that table to the Supplemental Material, page 14.

Reviewer #2, Comment 7: Page 10, below Table I: "has been widely been characterized"

Author’s Response: Thank you very much for noticing this. We have corrected this typo.

Reviewer #2, Comment 8: Fig. 3. The position of the labels (a), (b), (c) is a bit confusing. Also, in the caption, is the sentence "moves the imaginary part of internal modes farther away from the y-axis" correct?. Finally, the caption mention the case "increase of inertia and reduction of damping" but the label 5 of the figure says "reduced inertia, reduced damping".

Author's Response: Thank you very much for noticing this. We have carefully reviewed the captions and corrected them all in the revised main manuscript.

Reviewer #2, Comment 9: In the discussion section, what do the author exactly mean with the sentence "but their dynamic response may violate the necessary industrial control and operation standards"? Please elaborate.

Author's Response: This sentence suggests that the changing dynamics of power grids in practice are also influenced by the presence of power system protection schemes, which aim to correct potential stability issues before their magnitude is large enough to cause problems at the level of the entire system. However, according to current standards, some of the behavior seen in this work might induce undesired operation of protective equipment that have to abide by the current grid codes (which were designed with synchronous generation in mind), as explained below in more detail.

We refer to the fact that the changing nature of inertia and damping in power systems across the world may alter systems dynamics and behavior; they can become faster, moving into the range of signals which are conventionally perceived as anomalous and due to a fault or outage. The conventional understanding of power systems though has led to the development of various grid codes and operation standards that are enforced around the world, specifically those related to frequency response. Three industry standards are mainly pertinent to the topic addressed in this paper.

The first standard is related to Automatic Underfrequency Load Shedding (UFLS). This code, for instance in the North American grid, classified as the PRC-006 standard, defines the limit at which the load should be shed if the frequency drops below a threshold. It takes a wide variety of variables into account including tripping points, duration of out-nominal frequency, intentional delays, etc. It also lays out the procedures to follow for compliance check and testing.

The second standard is related to the frequency response obligation. This code, for instance in the North American grid, is classified as the BAL-003 standard, defines frequency response obligation and frequency bias settings for generators. The settings for frequency response are set by balancing authorities and often to comply and meet the standard, the generation entities are prohibited (by the generation interconnection agreement) from changes or adjustments made to droop settings.

The third standard is related to relays that activate on the basis of rate of change of frequency (ROCOF). This code, for instance in the Great Britain (GB) grid, classified as the ROCOF-based loss of mains (LOM) protection settings, defines the procedure to disconnect inverter-based resources during an islanding event.

The operation of these relays uses the ROCOF value, and the changing nature of frequency response with higher shares of IBR challenges the operation of these relays.

In our paper we imply that the underlying factors that formed these standards have been understood on the basis of the synchronous machine-dominated grid and its associated behavior and dynamics. Knowing how behavior of the grid will naturally change with higher shares of IBR, particularly GFM, without necessarily becoming unstable, these variables and their compliance procedures mostly like require a reconsideration to prevent cascading failures by disconnection of components when they are needed the most. For example, the massive power blackout in London in 2019 was attributed to the maloperation of the ROCOF relays and subsequent disconnection of a wide number of distributed generation units, as an investigation led by the UK Office of Gas and Electricity Markets uncovered [R1]. Such a scenario was predicted in a study conducted by the University of Strathclyde and had made a recommendation to change the relay settings from the then legacy value of 0.125 Hz/s to 0.5 Hz/s or 1 Hz/s [R2]. This blackout incident led to change of the ROCOF threshold to 1 Hz/s with a time delay of 0.5 second [R3].

We have added a more detailed description in the revised manuscript, main document, and elaborated on a few specific industrial control and operation standards. In the revised paper, it reads as follows, in Page 15 of the Main manuscript:

“Our theoretical and numerical results prove that the power networks with 100% renewable generation possess stable synchronization measures for dealing with small-signal perturbations, but their dynamic response may violate the necessary industrial control and operation standards; i.e., under frequency load-shedding (UFLS) scheme, the standards set for rate of change of frequency (ROCOF)-based relays, and frequency response obligation set by the balancing authorities.”

And in Page 14 of the Main document, it now reads

“We emphasize here that a stable case is a network whose response is bounded, and thus analytically stable in the sense of Lyapunov's first theorem and not in the sense of practice. In practice, the conformance to operational standards and procedures could introduce many non-linear elements such as under-frequency load shedding, frequency ride-through, etc.”

[R1] “9 August 2019 power outage report”, UK Office of Gas and Electricity Markets, Available [online]: https://www.ofgem.gov.uk/sites/default/files/docs/2020/01/9_august_2019_power_outage_report.pdf

[R2] “Assessment of Risks Resulting from the Adjustment of ROCOF Based Loss of Mains Protection Settings”, University of Strathclyde, 2015 Available [online]: <https://www.nationalgrid.com/sites/default/files/documents/Appendix%201%20Strathclyde%20Report%201.pdf>

Response to Reviewer #3

Reviewer #3, Comment 1: The article “Inherent Synchronization in Electric Power Networks with near 100% Inverter-Based Renewable Generation” discusses conditions of synchronization of power grids and claims that such an operation is possible with 100% inverter-based renewable generation.

The authors investigate how a power grid, in particular, one with heterogeneous parameters and potentially a high share of inverters can synchronize and thereby run stably without any added control action.

The topic is of great interest for an interdisciplinary audience and the quality of the figures and text is high. I feel some claims made by the authors are too strong based on the results presented and the presentation can be improved.

Overall, I recommend publication after revision.

Author’s Response: We sincerely appreciate your recognition of the merits of our paper.

Reviewer #3, Comment 2. The title suggests a different focus than the abstract and then the actual results: Both title and abstract should reflect that the focus is on synchronization criteria and stability considerations of networks with heterogenous parameters and potentially grids that are based 100% on inverter-based generators.

Author’s Response: Thank you very much for your insightful recommendation. Accordingly, we have updated the title and abstract to highlight the importance of heterogenous parameters and potentially grids that are based 100% on inverter-based generators. The new title is

Synchronization in Electric Power Networks with Inherent Heterogeneity Up to 100% Inverter-Based Renewable Generation

And the abstract reads as follows:

The synchronized operation of power generators is the foundation of electric power network stability and a key to the prevention of undesired power outages and blackouts. Here, we derive the conditions that guarantee synchronization in power networks with inherent generator heterogeneity when subjected to small perturbations and perform a parametric sensitivity analysis to understand synchronization with varied types of generators. As inverter-based resources, which are the primary interfacing technology for renewable sources of energy, have supplanted synchronous generators, the center of attention on associated integration challenges have resided primarily on the critical role of system inertia. Our results instead highlight the critical role of generator damping in achieving this pivotal state. Additionally, we report the feasibility of operating interconnected electric grids with up to 100% power contribution from

inverter-based renewable generation technologies. Our study has important implications as it sets the basis for the development of advanced control architectures and grid optimization methods and has the potential to further pave the path towards the decarbonization of the electric power sector.

Reviewer #3, Comment 3: I am not convinced that the last paragraph in the introduction is necessary. Instead, I would have expected some overview of the paper and its structure.

Author's Response: Thank you for pointing that out. In the revised paper, we have removed that last paragraph. However, a quick review of *Nature Communications* publications on the similar topic of power grids suggested that having a paragraph about the structure of paper does not match the common paper structure for *Nature Communications*, though in other publications venues, such as *Elsevier*, *IEEE*, and *IET*, it is common practice. Therefore, we simply removed that last paragraph.

Reviewer #3, Comment 4: What data was used to create Fig1? Did the authors themselves have PMUs in the CE grid?

Author's Response: That is correct. We have the original PMU data from the CE grid and used it in this paper.

Reviewer #3, Comment 5: On page 5 the authors implicitly utilize the DC approximation as they state the transmission line dynamics is governed by the angle difference (instead of the sine of the angle difference). Not even in the methods do the authors mention the term "DC approximation".

Author's Response: Thanks for pointing that out. We updated the text in the main manuscript mentioning that by DC approximation, the mechanical and the electrical analogs form a similar expression. Though, for the remainder of the paper, we used the complete power flow expressions with sine and cosine functions. On page 5 of the Main manuscript we used the DC approximation only for comparison purposes between the mechanical and electrical analogs. Everywhere else in the manuscript we utilize the AC formulation.

Reviewer #3, Comment 6: Follow-up of point 1: Only around pages 5-6 the authors indicate that heterogeneous parameters are critical. Why didn't they mention this point already in the title, abstract, introduction?

Author's Response: We have revised the title, abstract, and introduction and have highlighted the criticality heterogeneous parameters. We hope our revisions meets the esteemed reviewer's expectations.

Reviewer #3, Comment 7: When discussing equation (2): The authors include a “damping term” but explicitly exclude automatic control. It seems primary/droop control is used and it should be mentioned in the discussion of the results that increasing damping (or in this case droop control) is stabilizing for the system. Since droop control is adding power to the system this result is expected.

Author’s Response: We agree. We have updated the discussion in the paper and state explicitly that the “damping” term considered in our generic model takes into account all damping components including the damping torque for SG, droop control for SG and GFM, governor response and turbine time in SG, power measurement low-pass filter in GFM, and grid-coupling filter in GFM. Therefore, the title also has been updated to exclude the “inherent” term.

The take away message here is that the damping component of frequency response in synchronous machines is more restrictive than in inverters; both because of slower mechanical response and also the fact that the majority of control settings are mechanical built-in features of the machine and bounded by its physics and, therefore, cannot really be adjusted in real-time. Whereas the inverter-based generators can be utilized such that their droop value is adjusted in real-time and therefore the damping becomes an adaptive function.

To highlight this model, in the revised Supplemental Material, we have added a Laplace block diagram of the control processes that form the aggregated model we used for synchronous generator and the multi-loop grid-forming technology, as shown below, in Fig. R1.

(i) Synchronous generator

(ii) Multi-loop grid-forming inverter

Fig. R1. Laplace block diagram of the aggregated model for frequency response

We verified and validated these reduced-order models for studying frequency response using the full-order models in PSCAD, as shown in Fig. R2. The results show that these reduced order models are quite accurate in producing the SG and GFM frequency response.

— Full-order Model
 - - Reduced-order Model

(a) SG

— Full-order Model
 - - Reduced-order Model

(b) GFM

Fig. R2. Validation of the analytical model using full-order model implemented in an industry-grade power system planning software for electromagnetic transient modelling (EMT) dynamic modelling, per generation technology

Given the transfer function of both the SG and the GFM reduced-order models (those shown in the above Laplace block diagram) yield a second-order transfer function, next, we introduced a generic second-order model in the form of

$$\begin{aligned} \dot{x}_1 &= x_2 \\ \dot{x}_2 &= M^{-1}(P^* - P_e - Dx_2) \end{aligned}$$

that we used to represent both SG and GFM in which M and D , are the inertia and damping components that represent all contributing elements (the values of M and D vary for SG vs. GFM technology). We validated the generic models by comparing their resultant frequency response with those from the reduced order models presented in the Laplace diagram with explicit elements included (see Fig. R3 below).

- - Reduced-order Model
 — Generic Second-order Model

(a) SG

- - Reduced-order Model
 — Generic Second-order Model

(b) GFM

Fig. R3. Validation of the analytical generic second-order models by comparing the resultant frequency response with those from the reduced-order model (previously validated using the full-order model implemented PSCAD)

Reviewer #3, Comment 8: “Given our timescale of interest in this study, it is assumed that the load is constant.” What exactly is the time scale of interest here?

Author’s Response: This study is interested in cycles to seconds timescales and therefore, other phenomena such as seasonal load variation, very fast switching dynamics, or transmission line very fast dynamics, are not considered and all those values are assumed constant and hence modelled algebraically, all standard assumptions in power system dynamic studies. The only variation of load considered is the switching of one load in the studied scenario to perturb the system.

Reviewer #3, Comment 9: Equation (4) is confusing: Firstly, the difference between ω_i and ω_n is not clear. Secondly, the dimension of all entries should be indicated more clearly. In particular, the “0” entries to node have the same dimension but can be $1 \times (n-1)$ or $(n-1) \times (n-1)$ or $(n-1) \times 1$. The -1 matrix also seems to be $(n-1) \times 1$, based on Supplementary Equation (25). Please check the notation and make it easier to read and process.

Author’s Response: Thank you for pointing this out. We have carefully reviewed the notation in Equation (4) and updated the manuscript such that the notation dimensions and the notation themselves are clarified. Also, supplemental material Eq. (25) exhibits the extended form of Eq. (4) of the main manuscript in which the consistency of the dimensions of the submatrices is extended.

Reviewer #3, Comment 10: How can equation (5) as a third-order polynomial yield $2n-1$ solutions for λ ?

Author’s Response: Please note in the supplemental material, section S5, *Deriving the stability condition*, where we explain a technique that we have used to project the $(2n-1)$ th eigenvector into the other $2(n-1)$ dimensions only to find the solution in a convenient form of the characteristic matrix shown in Eq. (5) of the main manuscript. Because of this eigenvector projection, there will be $3(n-1)$ eigenvectors produced, but only $2n-1$ of them are linearly independent. The mathematical procedure used for this projection that produced Eq. (5) is explained in the supplemental material from Eq. (31) through Eq. (37) and then immediately we explicitly show this procedure for the 3-generator system as an example which follows from Eq. (38) through Eq. (48) of the Supplemental material, for better clarity.

Reviewer #3, Comment 11: The results in Table I seem suspicious: The authors claim that with only inverters and thereby much smaller damping and inertia the nadir and return times stay essentially identical? Also, the scenarios are not really introduced or discussed in the text. Potentially this table could even be moved to the Supplements as it does not seem to contain important results for the main text.

Author’s Response: Thank you for highlighting this. We have added more details about the introduction of the case considered here and details of the dynamic and network data used for producing these results, in the revised manuscript – the new data is included in the supplemental material. We also have moved the Table 1 to the Supplemental material, as advised by the esteemed reviewer.

We hope the additional information about the case and data satisfies the expectation of the esteemed reviewer.

Fig. R6. One-line diagram of the 3-generator, 9-bus system, and the dynamic and network data

TABLE II: Parameters for the cases presented in Fig. 4 of the Main Manuscript

Case	Inertia Coefficient	Damping Coefficient	Nadir	ROCOF
All SG/GFM-VSM	(1.9099, 0.9549, 0.3820)	(1.8335, 1.8335, 1.1001)	59.91	0.276
2/3rd GFL+1/3rd SG	(1.5915, 0.7958, 0.3183)	(1.5279, 1.5279, 0.9167)	59.73	0.376
All GFM-Droop	(0.0080, 0.0040, 0.0016)	(0.3361, 0.3361, 0.2017)	59.94	0.720

To reproduce the results shown in Figure 3, the data from All SG/GFM-VSM can be plugged into the model. The network data are available in Fig 10 of Supplemental material.

Reviewer #3, Comment 12: The main text should at least mention the nature of the “small perturbation” is this a perturbation in power, angle, frequency or some system parameter? Does it persist or is it only active at one time instance? Checking the methods and supplements, I still do not know how this “small perturbation” looks like.

Author’s Response: In the main manuscript, we expanded the methodology of perturbation where we describe the perturbation considered is in the form of angle perturbation to reflect the change of power on demand from generators. This perturbation emulates load switching and there is one perturbation per scenario.

One should note that after Kron reduction, all the load nodes are eliminated and the only way to emulate load switching is to perturb the generator angle directly, with the goal being to observe the subsequent frequency response.

In the revised manuscript, the Main document, Page 12, now reads as

“[We] measured the kinetic energy that the perturbation of generators electric angle (in the form of a step response) induce whose dissipation allows the system frequency to reach a steady state.”

And in Page 20 of Supplemental material, it reads as

“To perturb the system and emulate the load switching, we subjected the electric angle of generators, δ , to a step change that replicates a change of the loading condition and subsequently, we monitored the electric speed of all generators, ω . Because we applied Kron reduction which is a common technique in power network synchronization studies, the load nodes are eliminated and there is no other mean to directly switch loads, other than switching the δ of generators in a sustained fashion.”

Reviewer #3, Comment 13: The authors claim “this confirms the feasibility of operating electric power networks on 100% renewable based generation”. However, they have only shown that power grids can synchronize using inverter-based technology. While the authors claim that on the time scale they are interested in (and do not specify) the load will not change, renewable generation can easily change on the time scale of few seconds, minutes etc. None of the results presented here tackle the balancing problem of renewable generation. Hence, I urge the authors to make their claim more moderate and specific to fit what they have actually shown.

Author’s Response: We completely agree with the esteemed reviewer. The timescale of frequency response is cycles to seconds, as reflected in the results shown. We agree that this paper tackles a problem that is a part of a larger portfolio necessary to transition to 100% inverter-based operational capability, namely part of the “inverter problem”, but not any part of “the balancing problem” as laid out by P. Denholm et al [R1]. The power balancing issue however fall under different timescales of seconds to minutes and even years. Planners study that problem under resource adequacy in order to determine the capacity necessary to build years ahead, and the operators look into this problem under unit commitment for day(s) ahead and resource sufficiency among the balancing authorities to ensure the generation capacity, reserve margin, and ramping capability are available to meet the demand. Indeed, we did not intend to address all of the challenges of a transition to a 100% renewable grid, many topics are still open questions such as power limiting, fault detection and diagnosis, power sharing, ancillary service and market operation to hedge the risks associated with the variability and uncertainty inverter-based resources.

Overall, power systems have temporally distributed sets of decision-making that required multi-timescale measurement and operations. Figure R9 summarizes the multi-temporal dynamics of power system and highlights where in this spectrum our analysis falls.

Figure R9. Timescales in electric grid dynamics and operation, borrowed from [R2]

We updated the manuscript addressing this important topic and how our findings are only a piece of a larger portfolio. In the Discussion section, Page 15 of the main manuscript, we have added the following section.

“We recognized the solution established in this paper is an important part of a larger portfolio for a successful transition to decarbonized power networks. While we addressed the dynamics associated with frequency synchronization in timescale of cycles to seconds, the decarbonization of power networks involves challenges of varying timescale. On one end of the spectrum, it resides on power balancing assurance and resources adequacy that fall under timescales of minutes and even years, as planners study that problem under resource adequacy in order to find out the capacity necessary to build years ahead, and the operators look into this problem under unit commitment for day(s) ahead to ensure the generation capacity, reserve margin, and ramping capability are available to meet the demand. On this timescale, high shares of renewable technologies call for more flexibility of resources which many believe energy storage is the enabling technology. On the other end of spectrum, it resides on nonlinear transients that fall under timescales of milliseconds to cycles, as system is continuously susceptible to short-circuit faults and high frequency switching events. On this timescale, the high shares of renewable technologies require advanced prognosis, diagnosis, and isolation of impacted devices and cluster of the network. Across this spectrum, many more topics remain open questions such as power limiting, fault detection and diagnosis, power sharing, ancillary service, and market operation to hedge the risks associated with variability and uncertainty inverter-based resources.”

[R1] P. Denholm et al, “The challenges of achieving a 100% renewable electricity system in the United States,” *Joule*, Volume 5, Issue 6, 2021, Pages 1331-1352,

[R2] A. Von Meier, “Integration of renewable generation in California: Coordination challenges in time and space” Proceeding of 11th international conference on electrical power quality and utilisation (pp. 1-6). IEEE, October 2011

Reviewer #3, Comment 14: During the discussion the authors write “As a result, the control networks and structure for conventional power networks, which have been designed for a synchronous generator-dominated network to manage these machine unique dynamics, are likely inadequate for the operation of a 100% renewable generation-based grid because of the different underlying dynamics.” But how is this statement backed up by the results of this study? This seems like pure speculation.

Author’s Response: This statement stems from the fact that changing nature of inertia and damping in power systems alter systems dynamics and behavior; they can become more stiff or agile (manifesting itself in the form frequency response that declines either sharper or settles quicker – backed up by the results in this paper), a behavior that is conventionally perceived as anomalous, and thus requires protective action.

Conventional power systems have static protection schemes and presume that inverter-based resources are not responding (as almost universally all the currently installed inverters at the utility scale currently are grid-following). In the future power grid dominated by grid-forming inverters, new concepts such as adaptive protection that follows the real-time grid inertia and generators’ available headroom reserve or highly distributed control schemes can be utilized. Such ideas are of course adaptive to the new structure of the grid with high shares of grid-supporting IBRs. And that is what this sentence implies.

Accordingly, we added more discussion to support this statement in the revised manuscript. In the Discussion section, Page 15 of the main manuscript, we have added the following section.

“In future power networks dominated by grid-forming inverters, new concepts such as adaptive protection that follows the grid inertia to adjust its settings in real-time, and generators’ available headroom reserve are factored into the determination of the droop value in real-time, making the grid a dynamically adaptive network and to do so, various control schemes can be utilized, especially highly distributed control systems. Such ideas are of course adaptive to the new structure of the grid with high shares of grid-supporting IBRs.”

Reviewer #3, Comment 15: Connected to the previous point: How does the current paper act as a ” basis for the development of a control solution”? By giving the case without control so that further research can add control terms?

Author’s Response: The esteemed reviewer raises a very important point here. In our study, we have used aggregated models and not explicit control models. This is to show that if inverter-based resources are proactively participating in grid support functionalities as they have the capability to do, then how power grids are

controlled and operated should change. This could be as simple as implementing adaptive droop in real-time as a control strategy, which we show can really enhance grid stability, but would require changing protection schemes. Of course, our study is at a very high level, though it provides a proof of feasibility and sets the basis for granular control design that can be implemented in industry.

To further elaborate on this point, on Page 15 of the main manuscript, we have added the following sentence.

“While we have identified the stability improvements via adjustments to damping and inertial parameters, conventional power networks are equipped with control and protection apparatuses whose parameters are adjusted infrequently, if ever, if even possible (i.e., the mechanical inertia and damping of a synchronous generator are physical properties particular to a design). Furthermore, the majority of inverter-based resources integrated into the grid with the capability to adjust their parameters are often not obligated to provide frequency response support to the grid, because of both the technical capabilities (e.g., a lack of headroom reserve to respond) and the current real-time and ancillary services market structure (e.g., a lack of economic incentives or interconnection requirements).”

In the subsequent paragraph we then address how our study highlights the potential changes that can benefit the power industry in the future, as stated in the response to the previous comment:

“In the future power networks dominated by grid-forming inverters, new concepts such as adaptive protection that follows the grid inertia to adjust its settings in real-time, and generators’ available headroom reserve are factored into determination of the droop value in a real-time, making the grid a dynamically adaptive network and to do so, various control schemes can be utilized, especially highly distributed control systems. Such ideas are of course adaptive to the new structure of the grid with high shares of grid-supporting IBRs.”

Reviewer #3, Comment 16: There are a couple of typos/unclear expressions, e.g. “to damp out the transient forced induced in spring”, “GFM as applied in parallel on power systems an emerging and promising technology.” Especially in the captions, sentences tend to be very long and complicated. Please check and see whether they can be simplified.

Author’s Response: Thank you for pointing this out. We carefully reviewed the manuscript and have updated and tidied up the language. We hope our revisions meet the expectations of the esteemed reviewer.

Reviewer #3, Comment 17: Finally, I ask the authors to include a code and data availability statement and publish their code for transparency and reproducibility reasons in a publicly accessible way (e.g. github, OSF, zenodo...)

Author's Response: We indeed concur with the esteemed reviewer and are supportive of research reproducibility and transparency. We have pledged to the Associate Editor that we will make all the datasets that are used to produce the results presented in this paper, available to the public open source, if we receive a recommendation to publish in this journal from the esteemed reviewers, and of course, ahead of the final production of the paper. However, at this stage in the review process, to prevent the authors' identities from being revealed and comply with the double blind-review process policies, we are unable to do so.

REVIEWERS' COMMENTS

Reviewer #1 (Remarks to the Author):

All comments and questions of reviewer #1 have been answered by the authors on a high quality level.

Thus, reviewer #1 is satisfied with his part of the revision. Since the topic of the article is of great interest to an interdisciplinary audience and the figures as well as the text are of high quality, reviewer #1 recommends the publication of this paper after the complete revision has been carried out.

Prof. Dr.-Ing. H.P. Beck

Reviewer #2 (Remarks to the Author):

In the revised version of manuscript NCOMMS-21-45155 and the response to the referees the authors have extensively addressed all the points raised. In particular my main concern about the relation between inertia, damping, and primary and secondary frequency control has been clarified. The authors clearly explain now that their study addresses synchronization stability at timescales from a few hundreds of a second to seconds, faster than the frequency control timescales. With this regard the paper is sound and sheds some light on the stability problem of the power grid with reduced inertia, although limited to these particular timescale. The overall stability of the power system with respect to reduced inertia is, then, out of the scope of this work.

All the other points have also been satisfactorily addressed and, therefore I have no objection in that the paper is published in the present form. I still find the interest, novelty, and relevance of the results a little limited for Nature Communications. On a positive side I must admit that, even if well known, the analogy with a mechanical system might appeal a broader audience.

Reviewer #3 (Remarks to the Author):

I thank the authors for their very thorough revision and detailed replies to all the points raised by all reviewers. I recommend publication.

There are two remaining points I would like to comment on:

1. In the revised manuscript the abbreviation "GFM" is used before it is defined. The authors should carefully check that no further consistency issues arose from the new text pieces added to the manuscript.
2. I understand that the authors delay the publication of their code and data until the double-blind peer review has concluded. I still stand to my point that publication of this is critical for transparency and reproducibility and I urge the Editor to ensure that this is done.

Reviewer #1

All comments and questions of reviewer #1 have been answered by the authors on a high quality level. Thus, reviewer #1 is satisfied with his part of the revision. Since the topic of the article is of great interest to an interdisciplinary audience and the figures as well as the text are of high quality, reviewer #1 recommends the publication of this paper after the complete revision has been carried out.

Reviewer #2

In the revised version of manuscript NCOMMS-21-45155 and the response to the referees the authors have extensively addressed all the points raised. In particular my main concern about the relation between inertia, damping, and primary and secondary frequency control has been clarified. The authors clearly explain now that their study addresses synchronization stability at timescales from a few hundreds of a second to seconds, faster than the frequency control timescales. With this regard the paper is sound and sheds some light on the stability problem of the power grid with reduced inertia, although limited to these particular timescale. The overall stability of the power system with respect to reduced inertia is, then, out of the scope of this work.

All the other points have also been satisfactorily addressed and, therefore I have no objection in that the paper is published in the present form. I still find the interest, novelty, and relevance of the results a little limited for Nature Communications. On a positive side I must admit that, even if well known, the analogy with a mechanical system might appeal a broader audience.

Reviewer #3

I thank the authors for their very thorough revision and detailed replies to all the points raised by all reviewers. I recommend publication.

Author's Response: We sincerely appreciate all three esteemed reviewers for their time and effort to review our paper which provided us with a critically pivotal feedback and observations. And thank you for your recommendation of publication.

There are two remaining points I would like to comment on:

Reviewer #3, Comment 1. In the revised manuscript the abbreviation “GFM” is used before it is defined. The authors should carefully check that no further consistency issues arose from the new text pieces added to the manuscript.

Author's Response: We added a definition for grid-forming inverter (GFM) in the Introduction section.

Reviewer #3, Comment 2. I understand that the authors delay the publication of their code and data until the double-blind peer review has concluded. I still stand to my point that publication of this is critical for transparency and reproducibility and I urge the Editor to ensure that this is done.

Author's Response: As we had pledged throughout the review process, we have made all of the datasets and codes that are used to produce the results presented in this paper, available to the public open source. This is for further transparency and reproducibility of our results.

- The analytical codes and data are available open-source on Github: <https://github.com/ahsajadi/sync>
- The PSCAD models are also available open-source on Github <https://github.com/NREL/PyPSCAD>